



# Overview of the Meso-NH model version 5.4 and its applications

Christine Lac[1], Jean-Pierre Chaboureau[2], Valéry Masson[1], Jean-Pierre Pinty[2], Pierre Tulet[3],
Juan Escobar[2], Maud Leriche[2], Christelle Barthe[3], Benjamin Aouizerats[1], Clotilde Augros[1],
Pierre Aumond[1,a], Franck Auguste[4], Peter Bechtold[2,c], Sarah Berthet[2], Soline Bieilli[3], Frédéric Bosseur[5],
Olivier Caumont[1], Jean-Martial Cohard[2,b], Jeanne Colin[1], Fleur Couvreux[1], Joan Cuxart[1,d],
Gaëlle Delautier[1], Thibaut Dauhut[2], Véronique Ducrocq[1], Jean-Baptiste Filippi[5], Didier Gazen[2],
Olivier Geoffroy[1], François Gheusi[2], Rachel Honnert[1], Jean-Philippe Lafore[1], Cindy Lebeaupin
Brossier[1], Quentin Libois[1], Thibaut Lunet[4,e], Céline Mari[2], Tomislav Maric[1], Patrick Mascart[2],
Maxime Mogé[2], Gilles Molinié[2,b], Olivier Nuissier[1], Florian Pantillon[2], Philippe Peyrillé[1],
Julien Pergaud[1], Emilie Perraud[1], Joris Pianezze[3,6], Jean-Luc Redelsperger[6], Didier Ricard[1],
Evelyne Richard[2], Sébastien Riette[1], Quentin Rodier[1], Robert Schoetter[1], Léo Seyfried[2], Joël Stein[1,f],
Karsten Suhre[2,g,h], Marie Taufour[1], Odile Thouron[1], Sandra Turner[1], Antoine Verrelle[1], Benoît Vié[1],
Florian Visentin[1,i], Vincent Vionnet[1], and Philippe Wautelet[2]

[1]CNRM, Météo-France-CNRS, Toulouse, France
[2]Laboratoire d'Aérologie, Université de Toulouse, CNRS, UPS, Toulouse, France
[3]Laboratoire de l'Atmosphère et des Cyclones (LACy), UMR 8105 (Université de la Réunion, Météo-France, CNRS), Saint-Denis de La Réunion, France
[4]CERFACS, Université de Toulouse, CNRS, CECI, Toulouse, France
[5]Laboratoire SPE, Sciences Pour l'Environnement, CNRS, UMR 6134, Corte, France
[6]Laboratoire d'Océanographie Physique et Spatiale, UMR 6523 (Ifremer, IRD, UBO, CNRS), Brest, France
[a]now at: IFSTTAR, AME, LAE, F-44341 Bouguenais, France
[b]now at: Université Grenoble Alpes, Institut des Géosciences de l'Environnement, CNRS, CS 40 700, 38058 Grenoble Cedex 9, France
[c]now at: ECMWF, Reading, UK
[d]now at: University of the Balearic Islands, Palma, Mallorca, Spain
[e]now at: ISAE-SupAéro, Toulouse, France
[f]now at: DIROP/COMPAS, Météo-France, Toulouse, France
[g]now at: Institute of Bioinformatics and Systems Biology, Helmholtz Zentrum München, Neuherberg, Germany
[h]now at: Bioinformatics Core, Weill Cornell Medical College, Doha, Qatar
[i]now at: Revenue Canada Agency, Montréal, Canada

*Correspondence to:* Christine Lac (christine.lac@meteo.fr)

**Abstract.** This paper presents the Meso-NH model version 5.4. Meso-NH is an atmospheric non hydrostatic research model that is applied to a broad range of resolutions, from synoptic to turbulent scales, and is designed for studies of physics and chemistry. It is a limited-area model employing advanced numerical techniques, including monotonic advection schemes for scalar transport and fourth-order centered or odd-order WENO advection schemes for momentum. The model includes state-
5 of-the-art physics parameterization schemes that are important to represent convective-scale phenomena and turbulent eddies, as well as flows at larger scales. In addition, Meso-NH has been expanded to provide capabilities for a range of Earth system prediction applications such as chemistry and aerosols, electricity and lightning, hydrology, wildland fires, volcanic eruptions



and cyclones with ocean coupling. Here, we present the main innovations to the dynamics and physics of the code since the pioneer paper of Lafore et al. (1998) and provide an overview of recent applications and couplings.

## 1   Introduction

Since the 1990's, research-oriented models, such as MM5 (Fifth-Generation Mesoscale Model, Grell et al., 1995), WRF
(Weather Research and Forecasting, Skamarock and Klemp, 2008), Meso-NH (Lafore et al., 1998), and ARPS (Advanced Regional Prediction System, Xue et al., 2000, 2001), have played a crucial role in the advance of atmospheric studies. These models are powerful numerical laboratories that have been used to better understand atmospheric processes and to develop physical parameterizations of Global Climate models (GCMs) and Numerical Weather Prediction (NWP) models. They are also precursors of the convection-permitting numerical weather systems routinely operated since the late 2000's in the major
national weather services around the world and, more recently, of the convection-permitting models that are beginning to be used for regional climate simulations.

The Meso-NH model has been a major player in this research modeling community and is a comprehensive model available for meso-scale atmospheric studies. A characteristic feature of Meso-NH is that it covers a broad range of scales, from planetary waves to near-convective scales down to turbulence. This is possible via two-way grid-nesting and its versatile design as the
model can be used both as a Cloud Resolving Model (CRM) and a Large-Eddy Simulation (LES), in which most (up to 90 %) of the turbulence energy is resolved, as well as a Direct Numerical Simulation (DNS).

The Meso-NH LES facilities are used for both process studies and the development of new physical parameterizations of coarser resolution models. Meso-NH runs in the same way as an LES and Single Column Model (SCM) simulation, assuming that the entire LES domain corresponds to a single grid box of a coarser NWP or climate model. In addition to the num-
ber of points, the two runs differ in their 3D or 1D version of the turbulence scheme and the activated parameterization in SCM, as deep or shallow convection or as a cloud scheme. The LES allows the main coherent patterns to be resolved and the fine-scale variability to be characterized via Probability Density Functions (PDFs) to develop parameterizations, while the SCM configuration allows them to be validated. Initially, LESs were primarily used in constrained idealized configurations (homogeneous initial fields, cyclic lateral boundary conditions). However, now they also concern real case studies with
open boundary conditions, sometimes with a downscaling approach using grid-nesting techniques, providing spatio-temporal turbulence characteristics difficult to retrieve from measurements alone (Guichard and Couvreux, 2017).

In addition, the physical parameterizations of the convection-permitting NWP model AROME (Applications of Research to Operations at MEsoscale; Seity et al., 2011), running operationally at Météo-France since the end of 2008 (at 2.5 km horizontal resolution initially and now at 1.3 km resolution, Brousseau et al., 2016), are inherited from Meso-NH and the common
physical parameterization schemes continue to be jointly developed. This forms a virtuous circle of parameterization validation because AROME allows a daily verification of a large variety of meteorological situations, while Meso-NH runs with various configurations and resolutions including additional advanced diagnostics.



In addition to atmospheric studies, Meso-NH has been extensively used for various innovative applications in Earth system sciences, such as hydrology (e.g., Vincendon et al., 2009), oceanography (e.g., Lebeaupin Brossier et al., 2009), optical turbulence for astronomy (e.g., Masciadri et al., 2017), wildland fire (e.g., Filippi et al., 2011), and atmospheric electricity (e.g., Barthe et al., 2012b). Meso-NH is also an on-line atmospheric chemistry model, handling gas-phases (Tulet et al., 2003; Mari et al., 2004), aqueous chemistry (Leriche et al., 2013), aerosols (Tulet et al., 2006) and volcanic eruptions (Durand et al., 2014; Sivia et al., 2015). It integrates the chemistry and dynamics simultaneously at each time step, which is essential for air quality and climate interactions, as shown by Baklanov et al. (2014).

Lafore et al. (1998) provided a general description of an early version of Meso-NH developed in the 1990's. Since then, the model code has significantly evolved and grown, including advanced numerical schemes with higher-order numerical accuracy and scalar conservation properties, a complete set of sophisticated physical parameterizations, an externalized surface, on-line coupling with chemical, aerosols and electricity schemes, and elaborate diagnostics. These notable changes result in more efficient simulations with higher stability and accuracy, being used on a broader range of topics. It is now a fast and highly parallel code (Jabouille et al., 1999) able to run on computers with more than 100,000 cores. This is indeed a key requirement to be able to perform LESs over large-grid domains (Dauhut et al., 2015). The Meso-NH code has been open access since its version 5.1, and a comprehensive scientific and technical documentation is available on the Meso-NH web site (mesonh.aero.obs-mip.fr). All these advances have made Meso-NH an attractive community model that is currently used in research institutes around the world. The model has also participated in a number of intercomparison studies (Chaboureau et al., 2016; Field et al., 2017, among the most recent examples). In addition, a total of 481 papers and 148 PhD thesis have been published by Meso-NH users.

The objective of this paper is to present the main model developments since Lafore et al. (1998)'s model description paper. The outline of the paper is as follows. First, a thorough description of the current version of the code (version 5.4) is given in Section 2 and the new aspects of the dynamical core, numerical schemes, and physical parameterizations are described in Sections 3 and 4, respectively. Section 5 presents the chemical and aerosol schemes, and Sections 6 & 7 present the original in-line diagnostics and couplings. A brief review of the model evaluation is included in Section 8. Future plans are introduced in Section 9 prior to the concluding remarks.

## 2 Model overview

### 2.1 Main characteristics

Meso-NH is a French mesoscale meteorological research model, initially developed by the Centre National de Recherches Météorologiques (CNRM - CNRS/Météo-France) and the Laboratoire d'Aérologie (LA - UPS/CNRS). It is a gridpoint limited area model based on a non-hydrostatic system of equations. The equations are written on the conformal plane to take into account the Earth's sphericity. Enforcing the anelastic continuity equation requires solving an elliptic equation with high accuracy to determine the pressure perturbation. Lafore et al. (1998) presented the classical Richardson iterative method. A





more efficient method following Skamarock et al. (1997) has since been developed, based on a conjugate-residual algorithm accelerated by a flat Laplacian preconditionner, and has been vertically and horizontally parallelized.

The model can run real cases or idealized cases, when some simplifications are introduced (e.g., simple orography or neglecting the Earth's curvature). It can be used in 3D, 2D or 1D form: the 2D and 1D forms are obtained by imposing an idealized

configuration and omitting the advection terms (in the transverse direction for 2D and in all three directions for 1D). The prognostic variables are the three velocity components $(u, v, w)$, the potential temperature $\theta$, the mixing ratios of up to seven categories of species, including vapor ($r_v$), cloud droplets ($r_c$), raindrops ($r_r$), ice crystals ($r_i$), snow ($r_s$), graupel ($r_g$) and hail ($r_h$), the subgrid turbulent kinetic energy (TKE), and additional reactive and passive scalars, including the hydrometeor concentrations from two-moment microphysical schemes.

Even though large grids are increasingly used with massively parallel computers (e.g., Pantillon et al., 2013; Dauhut et al., 2015), grid-nesting remains an efficient technique to take into account scale interactions, even for LES (Verrelle et al., 2017). Two-way interactive grid nesting has been implemented in Meso-NH according to Clark and Farley (1984) and is presented in Stein et al. (2000). This allows the simultaneous running of several models (up to eight) of different horizontal resolutions because the nesting is only applied horizontally. The downscaling flow consists of using the coarse mesh values (of the "fa-

ther"model) as boundary conditions for the fine mesh domain (the "son"), while the upscaling flow relaxes the coarse mesh fields towards the fine mesh spatial average on the coarse grid size in the overlapping area. The relaxation coefficient is set to $1/4\Delta t$ where $\Delta t$ is the coarse mesh model time step. The fields involved are the prognostic variables and the 2D surface precipitating fields to maintain consistency between the soil moisture of the two nested models.

## 2.2   The Meso-NH software

Meso-NH is maintained by computer and research scientists from LA and CNRM.  The code is written in Fortran 90. Running scripts are in shell and use makefiles. Much of the Meso-NH model has been parallel since 1999 (Jabouille et al., 1999). The domain decomposition is 2D, i.e., the physical domain is split into horizontal subdomains in the $x$ and $y$ directions, and the communication between multiple processes is achieved via the Message Passing Interface (MPI). In 2011, it was necessary to extend the model parallel capabilities to new computers, e.g., the first PRACE (Partnership for Advanced Computing in

Europe) petaflop computer, on issues concerning the I/O and the pressure solver. As a result, a sustained performance of 4 TFLOPS (tera floating-point operations per second) was obtained using a grid with 500 million points (Pantillon et al., 2011).

Meso-NH can adapt to most machine architectures from Linux PCs or clusters to Macs or supercomputers with an excellent scalability. Figure 1 shows the results obtained on MIRA, a Blue Gene/Q system at Argonne National Laboratory and HERMIT, a Cray XE6 at HLRS, the High Performance Computing Center Stuttgart. The sustained TFLOPS gradually increases with the

number of threads while remaining close to the optimal speedup. When using four OpenMP tasks instead of one, a speedup of more than 30 % can even be obtained. This results in a sustained performance of 60 TFLOPS using two billion threads.

The required libraries to run Meso-NH are NetCDF because the output files are in nc4 format, MPI and the GRIdded Binary (GRIB) Application Programming Interface (API) to use the European Centre for Medium-Range Weather Forecasts

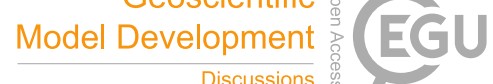



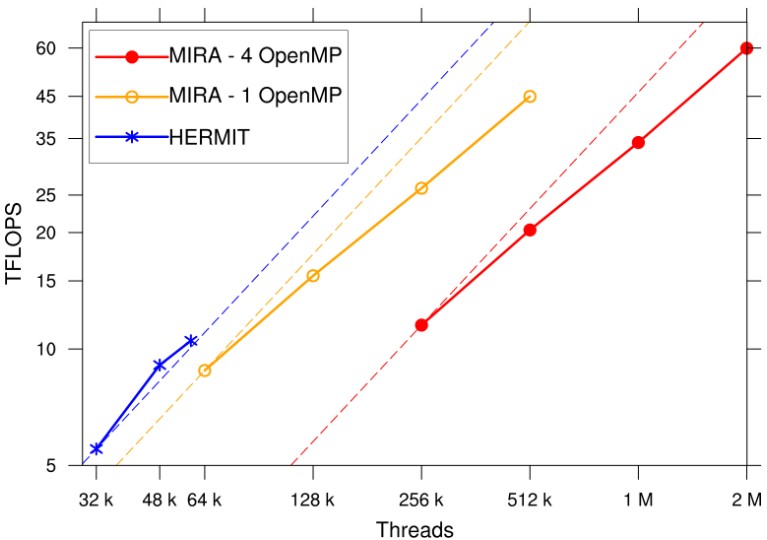

**Figure 1.** Performance of Meso-NH in scalability. Average sustained power (expressed in TFLOPS) depending on the number of threads obtained by Meso-NH for a grid of $4096 \times 4096 \times 1024$ points (17 billion points) on two machines (HERMIT, a Cray XE6 in Germany and MIRA, a IBM Blue Gene/Q in the USA using either one or four OpenMP tasks per core). The dashed lines show the optimal speedup.

(ECMWF) datasets. The code is bit reproducible, which means that the output fields are strictly the same for a given machine, regardless of the number of processors.

Meso-NH is also used for tutorials at the master level. The model can be easily installed and run on any computer, including small workstations or personal laptops. Furthermore, the model can be used under a two-dimensional framework allowing simulations to be obtained in only a few minutes. This makes Meso-NH a practical educational tool for studying numerical methods and atmospheric processes.

## 2.3 The code's organization

The Meso-NH framework is composed of three distinct blocks, running in a multi-tasking mode and corresponding to the following steps:

- the preparation step of a simulation where the user has to choose between the preparation of initial fields corresponding to idealized or real atmospheric conditions or the spawning of initial fields for a nested domain from initial or simulated fields of a father Meso-NH model;

- the temporal integration of the models, starting with the initialization step for each model and followed by the simulation integration of each model;

- the post-processing step to compute additional diagnostic fields.



A schematic overview of one integration time step of the model, with the different processes affecting the prognostic variables, is presented in Fig. 2. The time stepping is applied with a parallel splitting approach, meaning that all process tendencies are computed from the same model state and then the sum of the tendencies is used to step forward.

## 3  Dynamical core and numerical schemes

### 3.1  Governing equations

The dynamical core of Meso-NH solves the conservation equations of momentum, mass, humidity, scalar variables and the thermodynamic equation derived from the conservation of entropy under the anelastic approximation. The temperature, density and pressure are therefore described as small fluctuations from vertical reference profiles that are functions of height only. These equations are the same as in Lafore et al. (1998), in which further details can be found. The vertical coordinate is a height-based terrain-following coordinate. In addition to the originally implemented vertical coordinate (Gal-Chen and Somerville, 1975), it is also now possible to use the Smooth-Level Vertical-Coordinate (SLEVE) (Schär et al., 2002) where small-scale features in the coordinate surfaces decay rapidly with height, limiting the existence of steep coordinate surfaces to the lowermost few kilometers above the ground. For specific studies, it is possible to select a vertical domain that does not extend down to the ground, as in Paoli et al. (2014).

### 3.2  Transport schemes

Meso-NH is discretized on a staggered Arakawa C-grid, where meteorological variables (temperature, water substances, and turbulent kinetic energy) and scalar variables are located in the center of the grid cell and the momentum components are located on the faces of the cells. Due to the C-grid, the advection schemes are different for these two types of variables. The transport schemes consider the equations in their flux form to ensure conservation:

$$\frac{\partial}{\partial t}(\tilde{\rho}\phi) = -\frac{\partial}{\partial x}(\tilde{\rho}u_c\phi) - \frac{\partial}{\partial y}(\tilde{\rho}v_c\phi) - \frac{\partial}{\partial z}(\tilde{\rho}w_c\phi) \tag{1}$$

where $(x,y,z)$ are the transformed coordinates, $\tilde{\rho}$ is the dry density of the reference state, $\phi$ is the variable to be transported, including the wind components, and $(u_c, v_c, w_c)$ is the "advector" field, corresponding to the contravariant components, i.e. the components of the wind orthogonal to the coordinate lines, due to the conformed horizontal projection and terrain-following vertical coordinates. In the Cartesian framework, the metric terms exactly cancel and $u_c$, $v_c$ and $w_c$ are equal to $u$, $v$ and $w$. For the sake of simplicity, only the $x$-derivative term is considered hereafter:

$$\frac{\partial(\tilde{\rho}u_c\phi)}{\partial x} = \frac{\partial(F_C(\tilde{\rho}u_c)F(\phi))}{\partial x} \tag{2}$$

$F_C(\tilde{\rho}U_c)$ contains the topologic terms, which integrate the terrain transformations. The second flux $F(\phi)$ is calculated on the mesh point without considering terrain transformation, using the selected advection scheme.



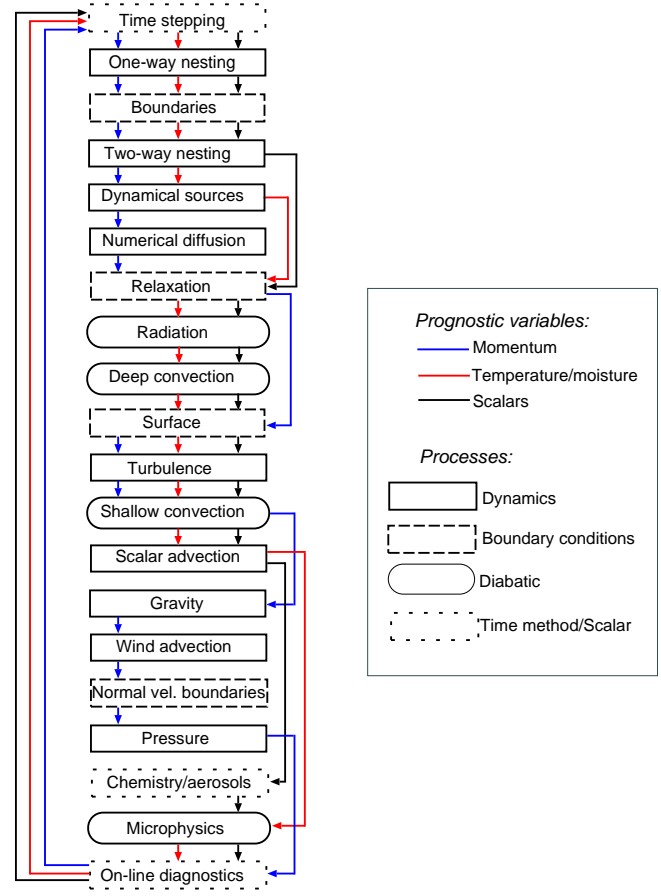

**Figure 2.** Flowchart of one integration time step of the simulation. The boxes represent the type of process, and the outline color represents the flows of the different types of variables.





The discrete form of the contravariant metric terms is second order in the horizontal directions and fourth order in the vertical direction in agreement with Klemp et al. (2003). The advection method for the wind variables and that for the scalars are distinct.

For the wind advection scheme, defining $i$ as the spatial index in the $x$-direction and $\Delta x$ the mesh size, the derivative is written such that:

$$\frac{\partial(\tilde{\rho}u_c u)_i}{\partial x} = \frac{F_C(\tilde{\rho}u_c)_{i+1/2}F(u)_{i+1/2}}{\Delta x} - \frac{F_C(\tilde{\rho}u_c)_{i-1/2}F(u)_{i-1/2}}{\Delta x} \tag{3}$$

Two different methods with distinct orders can be used to discretize $F$: a Weighted Essentially Non Oscillatory (WENO) discretization of fifth or third order (WENO5 and WENO3 respectively), or a centered discretization of fourth order (CEN4TH), as detailed in Lunet et al. (2017). WENO schemes owe their success to the use of an adaptive set of stencils, allowing a better representation of the solution in the presence of high gradients (Shu, 1998; Castro et al., 2011). The major asset of the fourth-order centered scheme is its good accuracy (effective resolution on the order of $5 - 6\Delta x$, Ricard et al., 2013).

The meteorological and scalar variable advection scheme is the Piecewise Parabolic Method (PPM), where piecewise continuous parabolas are fitted in each grid cell, enabling the scheme to handle sharp gradients and discontinuities very accurately. Three different versions of the PPM advection scheme have been implemented in Meso-NH: the unrestricted PPM_00, the monotonic version, PPM_01, based on the original Colella and Woodward (1984) scheme with monotonicity constraints modified by Lin and Rood (1996), and PPM_02, monotonic scheme with a flux limiter developed by Skamarock (2006). All three versions have excellent mass-conservation properties.

### 3.3 Time integration

A common strategy to improve computational efficiency is to use explicit time-splitting schemes as shown by Wicker and Skamarock (2002). In Meso-NH, Explicit Runge-Kutta (ERK) methods can be applied to the momentum transport, and Forward In Time (FIT) integration is applied to the rest of the model, including PPM and the contravariant flux $F_C(\tilde{\rho}u_c)$ transport. The different ERK methods are detailed in Lunet et al. (2017): the two main options are the fourth-order (RKC4) and the five-stage third-order (RK53) schemes.

To increase the maximum Courant-Friedrichs-Lewy (CFL) number, an additional time-splitting can be activated for the wind advection with WENO. One time step $[t_n, t_{n+1}]$ is divided into two regular sub-steps with a length of $\Delta t/2$. The intermediate tendencies are computed using all stages of the ERK method, and the final tendency is the half-sum of these two intermediate tendencies (Fig. 3a). The main interest of such an additional time splitting is to call the rest of the model (e.g., pressure solver, physics, and chemistry) less frequently: the larger time step is applied to the entire model including the physics and the pressure solver, with the FIT temporal scheme, while a smaller time step is used for the wind advection applying the ERK method on the subinterval. Lunet et al. (2017) have shown that such an additional two-time splitting results in an improvement of the maximum CFL number while a three-time splitting results in no further improvements.

CEN4TH can be applied with the RKC4 time marching (Fig. 3b) or with the leapfrog (LF) scheme, using in the latter case, the Asselin filter to damp the computational temporal mode.



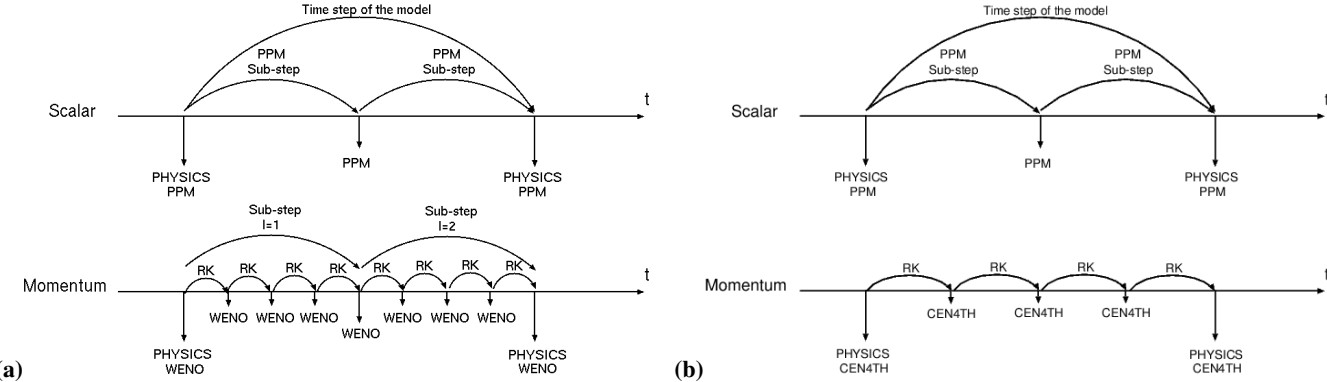

**Figure 3.** Representation of the time-marching in Meso-NH with (a) WENO5/RKC4 and (b) CEN4TH/RKC4 for the momentum transport.

An additional time-splitting can be activated for the scalar and meteorological variable advection to increase the time step of the rest of the model and to follow a CFL strictly less than 1 for the PPM (Fig. 3). This smaller time step for the PPM can evolve during the run as a function of the CFL number.

### 3.4 Numerical diffusion

The use of explicit numerical diffusion is prohibited with the PPM and WENO schemes. Only the fourth-order centered scheme for the momentum transport imposes a numerical diffusion operator for the wind to damp the numerical energy accumulation in the shortest wavelengths, with the RKC4 or LF time integration. The diffusion operator applied to the wind components $(u, v, w)$ is a fourth-order operator used everywhere except at the first interior grid point where a second operator is substituted in the case of non-periodic boundary conditions. Details can be found in Lunet et al. (2017). The user fixes the time at which

the $2\Delta x$ waves are damped by the factor $e^{-1}$.

Meso-NH can also be used to reproduce experiments - in hydraulic tanks and flumes - characterized by Reynolds number smaller than atmospheric ones by applying molecular diffusion to explicitly resolve the turbulence until the Kolmogorov scale (Gheusi et al., 2000). Viscous diffusion terms are added to the momentum and heat equations:

$$\frac{\partial}{\partial t}(\tilde{\rho}\boldsymbol{U}) = -\nu\nabla(\tilde{\rho}\nabla\boldsymbol{U}) \tag{4}$$

$$\frac{\partial}{\partial t}(\tilde{\rho}\theta) = -(\nu/P_r)\nabla(\tilde{\rho}\nabla\theta) \tag{5}$$

where $\boldsymbol{U}$ is the 3D air velocity, $P_r$ is the Prandtl number, defined as the ratio of the momentum diffusivity to the thermal diffusivity, and $\nu$ is the kinematic viscosity.

### 3.5 Comparison of the momentum and temporal schemes

Because various spatial and temporal schemes are available for momentum transport, their choice depends on the intended

use of the model and it is a compromise between the computing efficiency and the diffusive properties. A common method



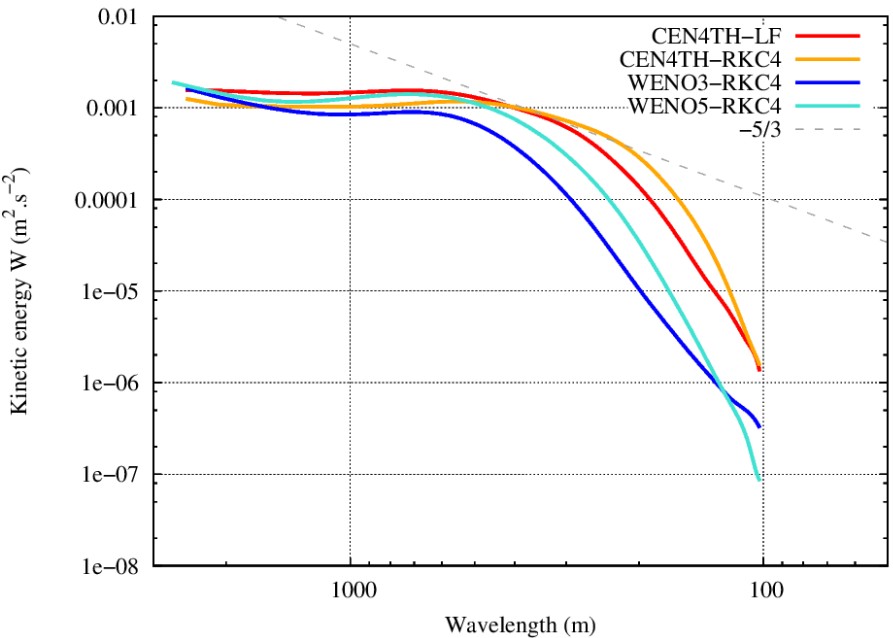

**Figure 4.** FIRE stratocumulus simulation case ($\Delta x = 50$ m) at 11 LT (local time) on 14 July 1987: mean kinetic energy spectra for the vertical wind computed in the boundary layer (between $0$ m and $1100$ m) with different numerical schemes for the wind transport. The dashed line indicates the power law with an exponent of $-5/3$ (the Kolmogorov spectrum).

to evaluate the diffusive behavior is to assess the effective resolution defined by the scale from which the slope of the model energy spectrum departs from the theoretical one (Skamarock, 2004; Ricard et al., 2013). Figure 4 displays the kinetic energy spectra for the FIRE stratocumulus case at a resolution of $\Delta x = 50$ m for the spatial and temporal schemes available in Meso-NH. It shows that CEN4TH/RKC4 presents a remarkable effective resolution (on the order of $4\Delta x$), followed by CEN4TH/LF

5  ($\sim 6\Delta x$), then WENO5/RK53-RKC4 ($\sim 8\Delta x$), the most diffusive being WENO3 ($\sim 10\Delta x$). Mazoyer et al. (2017) found similar results for the fog case. Some recommendations for numerical schemes are summarized in Tab. 1. CEN4TH/RKC4 is recommended for LES of clouds because the entrainment of environmental air at the cloud edges is higher with CEN4TH/RKC4 due to lower implicit diffusion, whereas WENO3 is inappropriate because it is excessively damping. However, WENO3 presents the best wall-clock time to solution and is recommended for long climate simulations for which the turbulence and cloud

10  processes are fully parameterized. WENO5/RK53-RKC4 is well adapted to sharp gradients area (Lunet et al., 2017), e.g., in complex shock-obstacle interactions with the immersed boundary method and in mesoscale case studies. The RK53 and RKC4 temporal schemes associated with WENO5 produce similar results.



| Wind transport scheme | Temporal scheme | Applications |
|---|---|---|
| CEN4TH | RKC4 | LES |
| WENO3 | RK53 or RKC4 | Climate - Chemistry |
| WENO5 | RK53 or RKC4 | Mesoscale - Sharp gradients |

**Table 1.** Recommendations for the choice of wind transport and temporal schemes according to the applications.

### 3.6 Initial and boundary conditions

As a limited area model, Meso-NH requires atmospheric initial and boundary conditions. These supply what we call the large-scale (LS) fields, which are used to initialize the prognostic variables, to force them at lateral boundaries with time-evolving fields, to define the background diffusion operator, or to relax the prognostic fields laterally or vertically. For real case studies,

initial and coupling fields can be provided by analyses or forecasts from the following NWP suites: AROME, ARPEGE (Action de Recherche Petite Echelle Grande Echelle), ECMWF and recently GFS (Global Forecast System). Initialization from ECMWF reanalyses is also possible. For ideal case studies, an initial vertical profile usually derived from observed radiosounding data can be provided by the user to be interpolated horizontally and vertically onto the Meso-NH grid to serve as initial and LS fields. The different forcing methods classically used in model intercomparison exercises, from geostrophic

winds to large-scale thermodynamical tendencies, are implemented in the code. Mostly used for long duration simulations, a nudging of the wind components, potential temperature and vapor mixing ratio towards the LS fields can be applied. In addition, an attribution method of filtering and bogussing has been introduced to the Meso-NH code to replace an ill-defined vortex in a large-scale field (Nuissier et al., 2005) or to isolate individual features from an ambient flow for further investigation (Pantillon et al., 2013). This method (Nuissier et al., 2005) consists of first filtering the large-scale fields of the wind, temperature and

humidity following the approach of Kurihara et al. (1993) and then adding the studied features or vortex to the likely filtered environmental conditions deduced from observations.

The lateral boundary conditions can be cyclic, rigid-wall or open and are detailed in Lafore et al. (1998). One change from the reference paper concerns the Carpenter method applied to the normal velocity component $u_n$:

$$\frac{\partial u_n}{\partial t} = \left(\frac{\partial u_n}{\partial t}\right)_{LS} - C^* \left(\frac{\partial u_n}{\partial x} - \left(\frac{\partial u_n}{\partial x}\right)_{LS}\right) - K\left(u_n - u_{n\,LS}\right), \tag{6}$$

where $C^*$ denotes the phase speed of the perturbation field $u_n - (u_n)_{LS}$ and is equal to $C^* = u_n + C$. To avoid eventual spurious waves at the lateral edges, $C$ is currently equal to 0 in the Planetary Boundary Layer (PBL), and to a constant adjustable phase speed in the free troposphere ($20\,\mathrm{m\,s}^{-1}$ by default). $K$ is usually set to $1/10\Delta t$. Another change is that, at the inflow boundaries, the scalar variables are interpolated between the large-scale and the interior values with a greater weight for the interior value (0.8), while they were taken to be the large-scale values in the reference paper.

The ceiling of the model is rigid corresponding to a free-slip condition. An absorbing layer can be added to prevent the reflection of gravity waves on this lid, where the prognostic variables are relaxed towards the LS fields. The bottom boundary



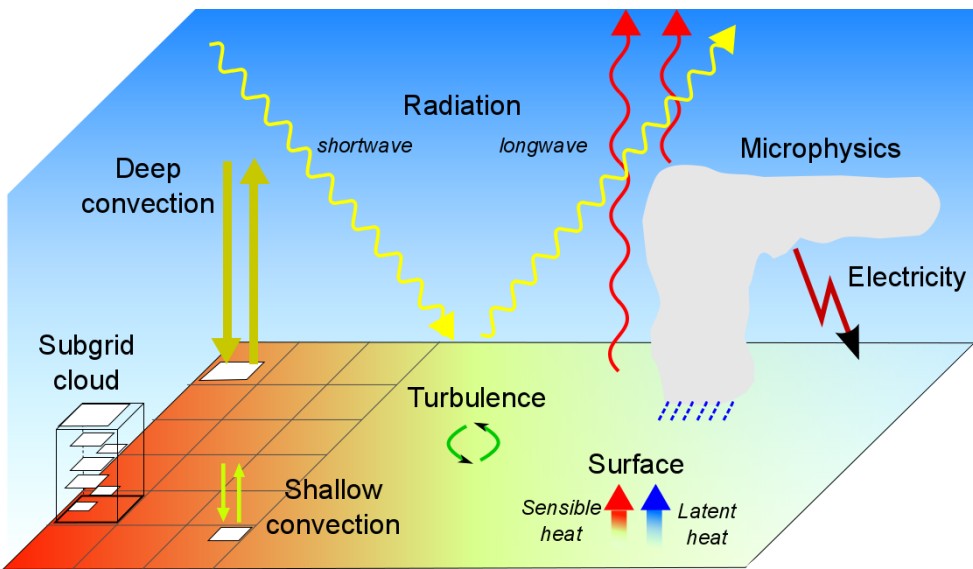

**Figure 5.** Physical parameterizations available in Meso-NH. The left-hand parameterizations are based on the implicit assumption because the processes they represent occupy only a portion of each grid mesh. The right-hand parameterizations represent several subgrid-scale processes that can be active over the full portion of each grid mesh.

considers a free-slip condition ($\overrightarrow{u}.\overrightarrow{n} = 0$). When performing DNS with Meso-NH, it is also possible to consider a no-slip bottom boundary condition ($\overrightarrow{u}(z=0) = \overrightarrow{0}$).

## 4 Physical parameterizations

In this section, a description of the physical parameterizations present in Meso-NH (Fig. 5) is given. We focus on the most
recent developments and some specific applications that are currently of great interest.

### 4.1 Surface

The surface schemes, initially available in Meso-NH, have been externalized to create SURFEX (Surface externalisée) standardized surface platform (Masson et al., 2013a); these schemes have been since enhanced by the contributions of different coupled models (from LES scale with Meso-NH to global climate simulation). Each grid box is split into four tiles: land, town,
sea and inland water (lakes and rivers). The main in-line schemes are the Interactions between Soil, Biosphere, and Atmosphere (ISBA) parameterization (Noilhan and Planton, 1989), the Town Energy Budget (TEB) scheme used for urban areas (Masson, 2000), and the freshwater lake model (FLake) used for lake surfaces (Mironov et al., 2010). Recently, a standard coupling interface was introduced to SURFEX (Voldoire et al., 2017) enabling coupling with various ocean and wave models to compute air-sea fluxes over the sea water tiles. The principle for the four tile types is that, during a Meso-NH time step, each surface
grid-box receives the potential temperature, vapor mixing ratio, horizontal wind components, pressure, total liquid and solid





precipitation, long-wave, short-wave and diffuse radiation, and possibly concentrations of chemical and aerosol species from the first atmospheric level above the ground. SURFEX returns the averaged fluxes for the sensible and latent heat, momentum, chemistry and aerosols, as well as the radiative surface temperature, surface direct and diffuse albedo and surface emissivity, which are used at the same first atmospheric level above the ground by the turbulence and radiation schemes. The coupling

method can be applied to any data flow between the soil and the atmosphere. Note that it is also possible to prescribe the energy fluxes and roughness length, possibly separately for each tile, to be able to perform theoretical studies, such as LES intercomparisons.

The vegetation scheme ISBA represents the effect of both vegetation and bare soil. The high vegetation can be simulated either as a separate layer above low vegetation, or as the more traditional and simplistic way of the 'big leaf' (all the vegetation being

then placed at ground level). Several evapotranspiration formulations are available for plants, the most advanced taking into account photosynthesis, respiration, and plant growth, and being able to simulate $CO_2$ fluxes as well. The soil is described either as a bucket of 2 or 3 layers or with a discretization in many (typically 14) layers, in which a root profile is defined. Freezing of the soil water is simulated, as well as snow mantel, with various degrees of complexity (the most complex snow scheme having many snow layers and simulating the evolution of the macro and micro physical characteristics of the snow).

The land tile can be separated in up to 19 subtiles, defined by the Plant Functional types, in order to perform more accurate vegetation and soil simulations, especially when photosynthesis and plant growth is simulated.

In order to keep the key processes governing the energy exchanges between the city and the atmosphere, the TEB scheme approximates the real city 3D structure of all buildings by keeping this 3D information under the form of an urban canyon: the road and urban vegetation being bordered by two very long buildings. This allows to take into account the effects of shadows

and radiative trapping, which limit the nighttime cooling, and the larger heat storage in the urban fabric during the day due to the larger surface in contact with the atmosphere (which leads to the heat island effect). Urban vegetaion (parks and gardens, trees and green roofs) are also simulated, with the ISBA scheme included in the TEB tile, and water interception and snow mantels on roofs and roads are also considered. A building energy module allows to simulate the needs in domestic heating and air conditionning, and the impact of the subsequent on the atmosphere. Human behaviour, building's uses and building's

architecture influence these heat emissions in the model.

The FLake scheme models the structure of the mixed and stratified water layers within the lakes using an assumed parametric form of the temperature profile. The effect of the sediments layer below the water is also considered, as well as the ice (and snow) above the water.

For the exchanges over sea surfaces, the surface fluxes are parameterized for a wide range of wind and environmental con-

ditions, from low winds to hurricanes. There is the possibility to use a coupled 1D ocean model. The single column model takes into account the vertical mixing within the ocean, as well as radiation absorption and surface energy balance. Also, the coupling with a 3D model, more detailed in Sect 7.1, is done through SURFEX. It allows to add the advection processes and the sea currents, at different scales. A wave model can also be activated, further modifying the surface fluxes.

Meso-NH version 5.4 includes SURFEX version v8.1. For a standard use of Meso-NH with SURFEX, four datafiles are

needed for the orography, clay and sand soil textures, and land use from Ecoclimap (Faroux et al., 2013) and Ecoclimap





second-generation. Global databases at 300 m (land cover, plants functional types, urban local climate zones (Stewart and Oke, 2012), vegetation parameters as leaf area index) and 1 km resolution (soil composition, lake depths,...) are available on the Meso-NH website. All parameters can also be prescribed separately by the user, as can the surface fluxes in an idealized configuration.

## 4.2 Turbulence

The turbulence scheme is based on Redelsperger and Sommeria (1982, 1986) and implemented in Meso-NH according to Cuxart et al. (2000a).

The scheme is built on the diagnostic expressions of the second-order turbulent fluxes, using the two quasi-conservative variables first introduced by Betts (1973) and Deardorff (1976), the liquid-water potential temperature $\theta_l$ and the non-precipitating total water mixing ratio $r_t = r_v + r_c + r_i$:

$$\overline{u_i'\theta_l'} = -\frac{2}{3}\frac{L}{C_s}e^{\frac{1}{2}}\frac{\partial\overline{\theta_l}}{\partial x_i}\phi_i, \tag{7}$$

$$\overline{u_i'r_t'} = -\frac{2}{3}\frac{L}{C_h}e^{\frac{1}{2}}\frac{\partial\overline{r_t}}{\partial x_i}\psi_i, \tag{8}$$

$$\overline{u_i'u_j'} = \frac{2}{3}\delta_{ij}e - \frac{4}{15}\frac{L}{C_m}e^{\frac{1}{2}}\left(\frac{\partial\overline{u_i}}{\partial x_j} + \frac{\partial\overline{u_j}}{\partial x_i} - \frac{2}{3}\delta_{ij}\frac{\partial\overline{u_m}}{\partial x_m}\right) \tag{9}$$

$$\tag{10}$$

where the Einstein summation convention applies for subscripts $n$; $\delta_{ij}$ is the Kronecker delta tensor; $\phi_i$ and $\psi_i$ are stability functions; and $C_s$, $C_h$ and $C_m$ are constant. Bars and primes correspond to means and turbulent components, respectively.

It includes the prognostic equation of the subgrid turbulent kinetic energy $e$, closed by the mixing length $L$, the dissipation being proportional to the subgrid TKE:

$$\frac{\partial e}{\partial t} = -\frac{1}{\tilde{\rho}}\frac{\partial}{\partial x_j}(\tilde{\rho}e\overline{u_j}) - \overline{u_i'u_j'}\frac{\partial\overline{u_i}}{\partial x_j} + \frac{g}{\tilde{\theta_v}}\overline{u_3'\theta_v'} + \frac{1}{\tilde{\rho}}\frac{\partial}{\partial x_j}(C_{2m}\tilde{\rho}Le^{\frac{1}{2}}\frac{\partial e}{\partial x_j}) - C_\epsilon\frac{e^{\frac{3}{2}}}{L}$$

$u_i$ being the $i$th component of the velocity, $\theta_v$ the virtual potential temperature, $\tilde{\theta_v}$ the virtual potential temperature of the reference state, $g$ the gravitational acceleration, and $C_{2m}$ and $C_\epsilon$ are constants.

At mesoscale resolutions (horizontal mesh larger than 2 km), it can be assumed that the horizontal gradients and the horizontal turbulent fluxes are much smaller than their vertical counterparts: therefore, they are neglected (except for the advection of TKE) and the turbulence scheme is used in its 1D version (noted T1D), as in AROME (Seity et al., 2011). At finer resolution, the entire subgrid equation system in its 3D version is considered (noted T3D), allowing LESs on flat or heterogeneous terrains. In the same way, the mixing length is diagnosed differently in the mesoscale and LES modes. At coarse resolution (typically greater than 500 m), the mixing length is related to the distance an air parcel can travel upwards ($l_{up}$) and downwards ($l_{down}$), constrained between the ground and the thermal stratification (Bougeault and Lacarrère, 1989). However, this mixing length, first built and evaluated for convective boundary layers, is unrealistic in purely neutral conditions (the upward length goes to the model top). In neutral but also stable conditions, the vertical wind shear constitutes the only positive source of TKE and is





of primary importance to influence turbulent eddies. Rodier et al. (2017) proposed a buoyancy-shear combined mixing length, by adding a local vertical wind shear term to the non-local effect of the static stability.

The mixing length for Bougeault and Lacarrère (1989) and Rodier et al. (2017) is defined by:

$$L = \left[ \frac{(l_{up})^{-2/3} + (l_{down})^{-2/3}}{2} \right]^{-3/2} \tag{11}$$

The distances $l_{up}$ and $l_{down}$ are defined by:

$$\int_{z}^{z+l_{up}} [\frac{g}{\tilde{\theta_v}}(\theta(z') - \theta(z)) + C_0\sqrt{e}S(z')]dz' = e(z),$$

$$\int_{z-l_{down}}^{z} [\frac{g}{\tilde{\theta_v}}(\theta(z) - \theta(z')) + C_0\sqrt{e}S(z')]dz' = e(z), \tag{12}$$

with

$$S = \sqrt{(\frac{\partial \overline{u_i}}{\partial z})^2 + (\frac{\partial \overline{u_j}}{\partial z})^2} \tag{13}$$

Note that Bougeault and Lacarrère (1989) formulas correspond to $C_0 = 0$.

When used in T3D mode, the horizontal mixing lengths are equal to the vertical one. In LESs, the mixing length can be linked to the largest subgrid eddies which have the size of a nearly isotropic grid cell

$$L = (\Delta x \Delta y \Delta z)^{1/3} \tag{14}$$

With strong stratification, these eddies are smaller; therefore, a mixing length reduced by stratification according to Deardorff

(1980) is proposed:

$$L = \min[(\Delta x \Delta y \Delta z)^{1/3}, 0.76\sqrt{e/N^2}] \tag{15}$$

where $N$ is the Brunt-Vaïsälä frequency.

Near the ground, the length scales of the subgrid turbulence scheme are modified according to Redelsperger et al. (2001) to match the similarity laws and the free-stream model constants.

To better represent the flow dynamics near the ground in the presence of complex plant or urban canopies, LESs are now frequently performed with meter-scale vertical resolution. Classically, the influence of these elements on the dynamics is introduced by the surface scheme via a roughness approach. A more realistic method is the drag approach (Aumond et al., 2013) in which drag terms are added to the momentum and subgrid TKE equations as a function of the foliage density for plant canopies:

$$\frac{\partial \alpha}{\partial t}_{DRAG} = -C_d A_f(z)\alpha\sqrt{u^2 + v^2} \tag{16}$$

with $\alpha = u, v,$ or $e$, where $u$ and $v$ are the horizontal wind components, $C_d$ is the drag coefficient, and $A_f(z)$ is the canopy area density.





This approach has been successfully used by Bergot et al. (2015) and Mazoyer et al. (2017) to study the impact of surface heterogeneities on the life cycle of fog. This new parameterization now allows the use of LESs in real case frameworks (Sarrat et al., 2017).

Inside convective clouds, Verrelle et al. (2015) have shown that turbulent mixing is insufficient in the updraft core, espe-
cially at coarse resolution (2 km), leading to strong resolved vertical velocities, even though it is better in T3D than in T1D (Machado and Chaboureau, 2015). LESs of convective clouds have shown that thermodynamical counter-gradient structures are present in convective clouds, as they are in convective boundary layers, and cannot be intrinsically represented by the common eddy-diffusivity turbulence scheme at mesoscale (Verrelle et al., 2017). The same study succeeded in reproducing the counter-gradient structures and increasing the thermal production of the TKE with the approach proposed by Moeng (2014)
which parameterizes the vertical thermodynamical fluxes in terms of horizontal gradients of resolved variables. Conversely, the necessity to increase turbulence at the cloud edges remains an active field of research.

### 4.3   Convection and dry thermals

At horizontal resolutions coarser than 5 km, it is necessary to parameterize shallow and deep convective clouds. The convection scheme available in Meso-NH, called KFB, is based on Kain and Fritsch (1990) with some adaptations presented in Bechtold
et al. (2000).  However, for shallow cumuli, KFB is not efficient enough, and does not represent dry thermals. The mass flux formulation of convective mixing, proposed in the EDMF (Eddy Diffusivity Mass Flux) approach (Hourdin et al., 2002; Soares et al., 2004) addresses this issue and has been introduced by Pergaud et al. (2009) into Meso-NH. This formulation considers a single entraining/detraining rising parcel starting from the ground. The vertical velocity equation is given by:

$$w_u \frac{\partial w_u}{\partial z} = a B_u - b \epsilon w_u^2 \tag{17}$$

where $w_u$ is the vertical velocity inside the updraft, $B_u$ is the buoyancy, $\epsilon$ is the entrainment rate, and $a$ and $b$ are constants. Entrainment and detrainment rates in the dry updraft are given by

$$\epsilon_{dry} = \max \left[ 0, C_\epsilon \frac{B_u}{w_u^2} \right], \tag{18}$$

and

$$\delta_{dry} = \max \left[ \frac{1}{l_{up} - z}, C_\delta \frac{B_u}{w_u^2} \right], \tag{19}$$

where $C_\epsilon$ and $C_\delta$ are constants. Mass flux continuity is ensured at cloud base between the dry and moist parts of the updraft. In the moist part, entrainment and detrainment rates are derived from the buoyancy sorting approach of Kain and Fritsch (1990). The closure assumption is given by the updraft initialization at the surface.

This scheme, called PMMC09, is also used in AROME at resolutions of 2.5 km and now 1.3 km and has considerably improved the realism of the clouds and winds in the PBL as shown by Lac et al. (2008); Seity et al. (2011). A comparison
between PMMC09 and five other mass-flux schemes using the AROME framework on the five French metropolitan radio-sounding locations over one year in Riette and Lac (2016) demonstrated the good performance of this scheme, which was





characterized by the active transport of thermals. In convective situations, it is necessary to use a mass-flux scheme such as PMMC09 until 1 km–500 m horizontal grid spacing. However, in this range of grid spacing, PBL thermals may be partly resolved and partly subgrid because they are in the grey zone of turbulence (Honnert et al., 2011). Honnert et al. (2016) showed that the mass-flux scheme, in its original form, is too active at this range of resolution, preventing the production of resolved

structures, and proposed several modifications to adapt PMMC09 to the grey zone.

### 4.4 Microphysics

Different bulk microphysical schemes are available in Meso-NH that predict either one or two moments of the particle size distribution (PSD) for a limited number of liquid or solid water species. One-moment microphysical schemes predict the mass mixing ratio of some water species, and two-moment schemes predict both the mass mixing ratio and the number concentration

of some species.

The most commonly used one-moment scheme is the mixed ICE3 scheme (Caniaux et al., 1994; Pinty and Jabouille, 1998) including five water species (cloud droplets, raindrops, pristine ice crystals, snow/aggregates and graupel), coupled to a Kessler scheme for warm processes. Hail is considered either as a full sixth category (Lascaux et al., 2006) or as forming with graupel an extended class of heavily rimed ice species. ICE3 is included in this latter form in AROME (Seity et al., 2011). The

particle sizes for each category follow a generalized Gamma distribution, with the particular case of the exponential Marshall-Palmer distribution for the precipitating species. Power-law relationships allow the mass and fall speed to be linked to the particle diameters. Cloud species are also handled by the subgrid transport (turbulence and shallow convection with PMMC09). Numerous processes exchanging mass between species are presented in Lascaux et al. (2006). All the microphysical processes are computed independently of each other with a mass budget at each step to ensure conservation. Following the microphysics,

an implicit adjustment of the temperature, vapor, cloud and ice contents is performed in clouds with a strict saturation criterion.

The complete two-moment scheme in Meso-NH is the mixed phase LIMA (Liquid Ice Multiple Aerosols) scheme (Vié et al., 2016) which is consistent with ICE3 and with the two-moment warm microphysical scheme from Cohard and Pinty (2000a, b). In addition to the five water mixing ratios of ICE3, LIMA predicts the number concentration of the cloud droplets, raindrops, and pristine ice crystals. The strength of the scheme is that it includes a prognostic representation of the aerosol population,

which is represented by the superimposition of several aerosol modes, each mode being defined by its chemical composition, PSD, and ability to act either as CCN, Ice Freezing Nuclei (IFN), or coated IFN (aged IFN acting first as CCN and then as IFN) as a function of its solubility. As in ICE3, LIMA assumes a thermodynamical equilibrium between the water vapor and cloud droplets. However, in the cold phase, the prediction of the concentration of ice crystals leads to an explicit computation of the deposition and sublimation rates, allowing under/supersaturation over ice. The microphysical processes of ICE3 and LIMA are

summarized in Fig. 6. The names of the processes are given in Tab. 2.

A variant to this scheme has been introduced by Geoffroy et al. (2008) for low precipitating warm clouds producing drizzle, following Khairoutdinov and Kogan (2000). Instead of a diagnostic saturation adjustment for the warm phase, Thouron et al. (2012) proposed, for LESs of boundary layer (BL) clouds, a pseudo-prognostic approach for supersaturation to limit the droplet concentration production and to better represent cloud top supersaturation due to mixing between cloudy and clear air.





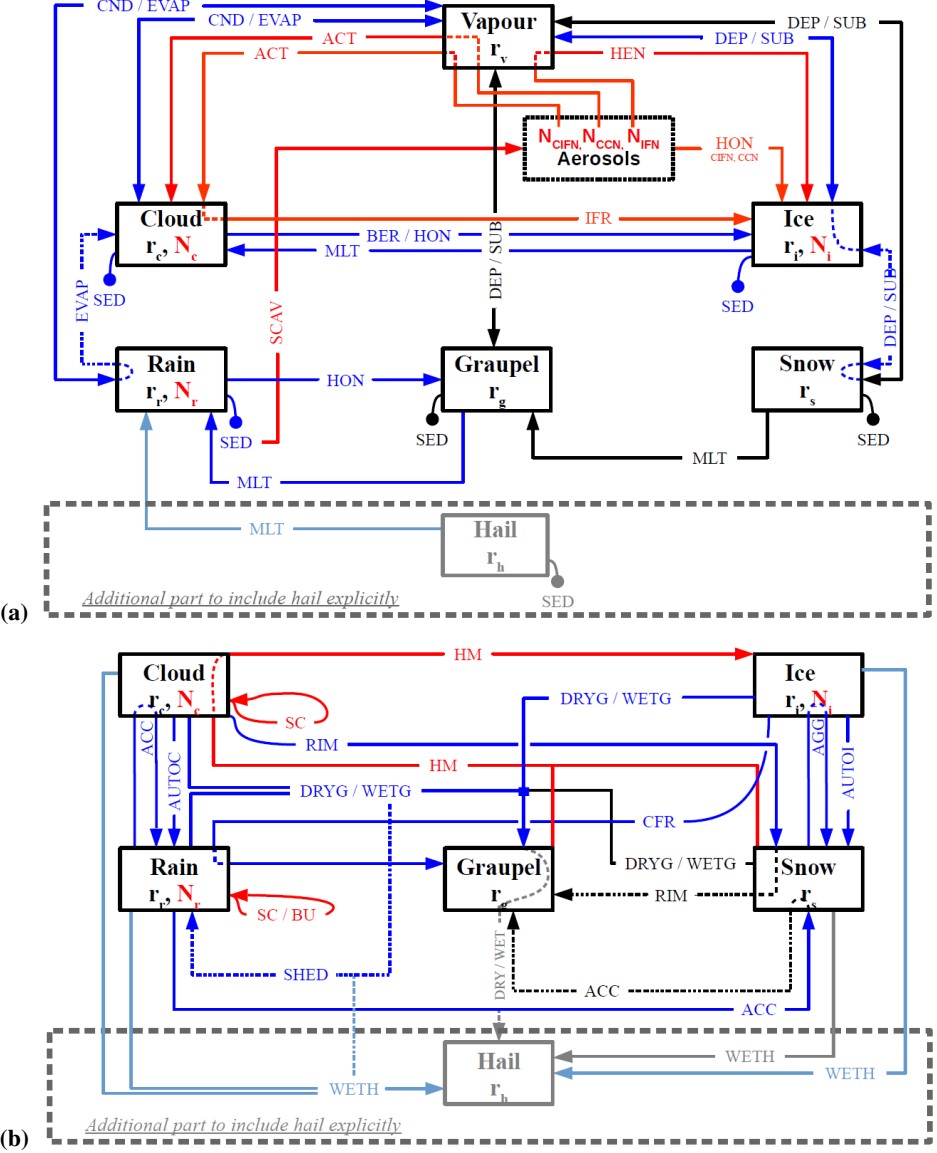

**Figure 6.** Diagrams of the microphysical processes of ICE3 and LIMA: (a) All the processes except collection; (b) Collection processes. Blue arrows represent existing processes in ICE3 modified in LIMA, red arrows are new processes in LIMA, and black arrows are identical processes in ICE3 and LIMA. When hail is a full sixth category, processes are in muted colours. Prognostic variables for all the hydrometeor species are written in the boxes, with $r$ the mixing ratio and $N$ the concentration.

The two-moment microphysical approach in Meso-NH has allowed numerous studies of the impact of aerosols on cloud life cycles to be conducted, e.g., for cumulus clouds (Pinty et al., 2001), stratocumulus clouds (Sandu et al., 2008, 2009), and fog (Stolaki et al., 2015).

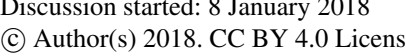
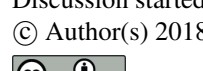

| Symbol | Process |
|---|---|
| ACC | Accretion (e.g. of droplets by rain drops) |
| ACT | CCN activation |
| AGG | Aggregation of pristine ice on snow |
| AUTOC | Autoconversion of cloud droplets into rain drops |
| AUTOI | Autoconversion of pristine ice crystals into snow |
| BER | Bergeron-Findeisen |
| CFR | Rain contact freezing |
| CND/EVAP | Condensation and evaporation |
| DEP/SUB | Deposition and sublimation |
| DRYG/WETG | Growth of graupel in the dry or wet regimes |
| HEN | Heterogeneous nucleation on IFN |
| HM | Hallett-Mossop |
| HON | Homogeneous freezing |
| IFR | Immersion freezing of coated IFN |
| MLT | Melting |
| RIM | Cloud droplet riming on snow |
| SC | Self collection of cloud droplets |
| SC/BU | Self collection and break-up of rain drops |
| SCAV | Below-cloud aerosol scavenging by rain |
| SED | Sedimentation |
| SHED | Water shedding |
| WETH | Growth of hail in the wet regime |

**Table 2.** List of the microphysical processes.

## 4.5 Subgrid cloud schemes

When the spatial resolution is not sufficient to consider the grid mesh completely clear or cloudy, a subgrid condensation
scheme can be activated with one-moment microphysical schemes, as suggested by Sommeria and Deardorff (1977) and Mellor
(1977), supplying a cloud fraction to the radiation scheme. The statistical cloud scheme is based on the computation of the
variance of the departure to the saturation inside the grid box, summarizing both the temperature and total water fluctuations.
PDFs of the saturation deficit are used to represent the statistical distribution of the cloud variability, and the cloud fraction
and mean cloud water mixing ratio can be deduced. A combination of unimodal Gaussian and skewed exponential PDFs is
defined for BL clouds according to Bougeault (1981, 1982). Chaboureau and Bechtold (2002, 2005) introduced the effects of
a deep convection scheme in the parameterization of the standard deviation of the saturation deficit. The subgrid variability
from the PMMC09 shallow convection scheme can be introduced in the same way via the variance of the saturation deficit, or



the cloud fraction can be diagnosed directly from the updraft fraction. The second method has been chosen for the operational version of AROME. Perraud et al. (2011) have conducted a statistical analysis with Meso-NH of LESs of warm BL clouds to show that double Gaussian distributions are more appropriate than unimodal theoretical PDFs when describing sparse subgrid clouds such as shallow cumuli and fractional stratocumuli, in agreement with Larson et al. (2001a, b) and Golaz et al. (2002a,

b). Because there can be other sources of subgrid variability, such as gravity waves in stable BLs, when the turbulence and shallow convective contributions are too weak to produce clouds, a variance proportional to the saturation total water specific humidity has been added, as in classical relative humidity cloud schemes (e.g., Rooy et al., 2010), and has shown significant improvement for winter clouds in AROME.

In the same way, a subgrid rain scheme has been developed by Turner et al. (2012) to simulate the gradual transition

from non-precipitating to fully precipitating model grids for warm clouds. A prescribed PDF of cloud water variability and a threshold value of the cloud mixing ratio for droplet collection are used to derive a rain fraction, and overlapping assumptions for the cloud and rain fraction are considered. In the future, this approach will be generalized to mixed microphysical processes and the PDFs between the subgrid cloud and rain schemes will be harmonized.

### 4.6   Radiation

Two radiation codes are available in Meso-NH, both originating from ECMWF and based on two-stream methods. The radiation code calculates the atmospheric heating rates and the net surface radiative forcing required to compute the temporal evolution of the potential temperature and the surface energy balance:

$$\frac{\partial \theta}{\partial t} = \frac{g}{C_{ph}} \Pi \frac{\partial F}{\partial p}, \qquad (20)$$

where $F$ is the net total flux: $F = F_{LW}^{\uparrow} + F_{LW}^{\downarrow} + F_{SW}^{\uparrow} + F_{SW}^{\downarrow}$ sum of the upward and downward shortwave (SW) and longwave

(LW) fluxes, and $C_{ph}$ the calorific capacity. In addition it returns as diagnostics the SW and LW fluxes at each model level in a number of spectral bands, distinguishing between the direct and diffuse components for SW. Clear sky quantities are also available. LW and SW radiative transfers are treated by distinct routines.

In the original code two LW radiation schemes were available: the Morcrette (1991) scheme based on an effective emissivity approach, composed of 9 spectral intervals, and the Rapid Radiation Transfer Model (RRTM, Mlawer et al., 1997) based on the

correlated k-distribution method, integrating 16 bands and 140 g-points (Morcrette, 2002). The SW radiation scheme applies the photon path distribution method employed by Fouquart and Bonnel (1980) in 6 spectral bands. The total cloud fraction is computed according to the cloud overlap assumption, and fluxes are calculated independently in the clear and cloudy portions before being aggregated.

The latest radiation code of ECMWF, ecRad (Hogan and Bozzo, 2016), was implemented in Meso-NH in 2017. This code is

highly modular, which allows the user to conveniently choose between multiple options. The main differences from the original code concern the implementation of the SW version of RRTM with 14 bands and 112 g-points (Morcrette et al., 2008) and some modifications regarding the treatment of unresolved cloud horizontal heterogeneities. The latter can now be treated with the McICA (Pincus et al., 2003) or TripleClouds (Shonk and Hogan, 2008) methods, or with the SPARTACUS solver (Schäfer





et al., 2016; Hogan et al., 2016), which represents lateral photon transport through the cloud sides (Hogan and Shonk, 2013) in a 1-D formalism. The overall code has also been rewritten, resulting in a 30 % reduction in the computation time compared to the original configuration. Aerosols are now prescribed via the mixing ratio vertical profiles of 12 different aerosol types corresponding to various physical properties and sizes according to CAMS (the Copernicus Atmosphere Monitoring Service, Stein et al., 2012). The optical properties of hydrophilic aerosols change with relative humidity, and their mixing ratios can be prognostic, or taken from the CAMS climatology (Bozzo et al., 2016), which replaces the former six-class climatology of Tegen et al. (1997) that used optical properties from Aouizerats et al. (2010).

In both radiative codes, liquid and ice cloud optical properties can be computed according to a variety of parameterizations. The liquid cloud optical radius is generally computed from the liquid water content following the parameterization of Martin et al. (1994) for the one-moment microphysical scheme, while it is deduced from the PSD in two-moment microphysics. Likewise, the ice cloud optical radius can be computed from the ice water content following Sun and Rikus (1999) and Sun (2001). Cloud optical properties (optical depth, single scattering albedo and asymmetry parameter) are then computed as a function of the particle effective radius following the parameterizations of Fouquart (1988) or Slingo (1989) for one-moment schemes, and Savijärvi et al. (1997) for two-moment schemes. Ice water optical properties can be computed according to Ebert and Curry (1993), Smith and Shi (1992), and Baran et al. (2014).

### 4.7 Electricity

Meso-NH is one of three CRMs having a completely explicit 3D electrical scheme. The scheme, called CELLS for the Cloud ELectrification and Lightning Scheme (Barthe et al., 2012a), computes the full life cycle of the electric charges from their generation to their neutralization via lightning flashes. An earlier version of this scheme (Molinié et al., 2002; Barthe et al., 2005) was gradually improved in order to cope with simulations of thousands of lightning flashes over large grids and complex terrain (Barthe et al., 2012a). It was developed from the one-moment bulk mixed-phase microphysics scheme ICE3 and its extension hail ICE4. The scheme follows the evolution of the mass charge density ($q_x$ in C kg$^{-1}$ of dry air) attached to each condensate species of the microphysics scheme:

$$\frac{\partial}{\partial t}(\tilde{\rho}q_x) + \nabla \cdot (\tilde{\rho}q_x \boldsymbol{U}) = \tilde{\rho}(S_x^q + T_x^q) \tag{21}$$

The source terms $S_x^q$ include the turbulence diffusion, the charging mechanism rates, the charge sedimentation by gravity and the charge neutralization by lightning flashes. $T_x^q$ is the transfer rates due to the microphysical evolution of the particles.

CELLS follows the positive and negative ion concentrations ($n_\pm$ in kg$^{-1}$), whose governing equation includes the drift in the electric field, the attachment to the charged hydrometeors, the release of ions when hydrometeors evaporate or sublimate, production via lightning flashes and via point discharge current from the surface, ion generation via cosmic rays, and ion-ion recombination. Fair weather conditions are computed following Helsdon and Farley (1987) and are used to initialize the positive and negative ion concentration profiles and to treat the lateral boundary conditions.

The cloud electrification is based upon the common assumption that the charge separation in thunderstorms mainly occurs during rebounding collisions between more or less rimed particles. However, there is still no consensus on the theory of so-





called non-inductive charging mechanisms. Therefore, several parameterizations of this process have been implemented into CELLS as described in Barthe et al. (2005). This set of parameterizations includes the well-known equations of Takahashi (1978), Saunders et al. (1991) and Saunders and Peck (1998), along with some improvements by Tsenova et al. (2013). The inductive process, which is efficient once an electric field is well established in the clouds, can also be activated (Barthe and

Pinty, 2007a). Electric charges are exchanged between hydrometeors during mass transfers due to microphysical processes. Each electric charge transfer rate is associated with a mass transfer rate in proportion to the electric charge density and inverse mixing ratio.

The electric field ($\overrightarrow{E}$) is computed from the Gauss equation forced by the total charge volume density ($\rho_{tot}$):

$$\nabla \cdot \overrightarrow{E} = \frac{\rho_{tot}}{\epsilon} \qquad (22)$$

with $\epsilon$ the dielectric constant of air. A pseudo electrical potential $V$ is introduced to convert the Gauss equation into an equivalent elliptic equation of pressure perturbations of Meso-NH:

$$\overrightarrow{E} = -\overrightarrow{\nabla}V \qquad (23)$$

$\overrightarrow{E}$ is then derived using a numerical gradient operator.

The lightning flash scheme was designed to reproduce the overall morphological characteristics of the flashes at the model

scale. Indeed, an accurate estimate of the lightning path would computationally be too expensive when simulating real meteorological cases over large domains (Barthe et al., 2012a). In order to treat several flashes in the same time step, an iterative algorithm was developed to identify and delineate all the electrified cells in the domain. A lightning flash is triggered once the electric field in an electrified cell reaches a threshold value ($E_{trig}$) that decreases with altitude as given by Marshall et al. (2005). In the first step, the flash propagates vertically as the bidirectional leader. In the second step, and to account for the hor-

izontal extension highlighted by very high frequency (VHF) mapping systems, a branching algorithm allows the 3D structure of the lightning flashes to be mimicked. As a result the grid point locations reached by the lightning "branches" are estimated according to a fractal law (Niemeyer et al., 1984).

The total charge in excess of $|0.1|$ nC kg$^{-1}$ is neutralized along the lightning channel. In the case of intra-cloud flashes, a charge correction is applied to all the flash grid points to ensure an exact electroneutrality prior to the redistribution of the net

charge to the charge carriers at the grid points. This constraint does not apply to cloud-to-ground discharges (charge leakage in the ground) which are defined when the tip of the downward branch of the leader reaches an altitude below 2 km above ground level. Once charge neutralization is completed, the electric field is updated. If a new triggering point is found in at least one of the detected cells, a new lightning flash is triggered. This allows several lightning flashes to occur during a single time step.

## 5 Chemistry and aerosols

Meso-NH integrates a complete set of processes to simulate changes in the atmospheric composition in terms of aerosols and trace gases from LES to continental scales. Initial and boundary conditions for gases and aerosols are processed following the

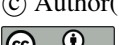



same procedure as the dynamical variables (Section 3.6). For real case studies, large-scale chemical fields are provided by two global models: Modèle de Chimie Atmosphérique à Grande Echelle (MOCAGE, Bousserez et al., 2007) and the Model for OZone And Related chemical Tracers (MOZART, Emmons et al., 2010). For ideal case studies, a user-prescribed horizontally homogeneous vertical profile is applied.

## 5.1 Emissions and dry deposition

The interactions of gases and aerosols with the surface are treated in the externalized surface model SURFEX (Section 4.1). Dry deposition processes commonly follow the resistance analogy described by Wesely (1989) and take into account the aerodynamic and canopy resistances as a function of land cover types and vegetation. A full description is given by Tulet et al. (2003). Dry deposition and sedimentation of aerosols are driven by Brownian diffusivity and the gravitational velocity. These processes are calculated over each mode of the aerosol size distribution (Tulet et al., 2005). For the sedimentation process, the gravitational velocity is solved using a time splitting technique to compute the sedimentation fluxes. Emissions for the model domain are complied from a prescribed emissions database or can be parameterized. The surface model can process the raw prescribed emission data from any inventory of primary gases or aerosols. Emissions can include urban and industrial, biogenic, biomass burning, and volcanic sources from the most recent emissions databases. Desert dust emissions are parameterized following the Dust Entrainment and Deposition model (DEAD, Zender et al., 2003) based on the pioneering work of Marticorena and Bergametti (1995). The dust emission scheme was incorporated into Meso-NH-SURFEX by Grini et al. (2006) and modified by Mokhtari et al. (2012) to better account for the size distribution of erodible material. Sea salt emission follows the parameterization of Ovadnevaite et al. (2014). Input parameters such as wind stress, significant wave height, salinity and sea surface temperature are taken from oceanic models such as CROCO (Coastal and Regional Ocean COmmunity model, Debreu et al., 2016) or NEMO (Nucleus for European Modelling of the Ocean, Madec, 2008) and from the wave model WW3 (WAVEWATCH III, Tolman et al., 2009). Biogenic emissions are either prescribed or calculated on-line based on the Model of Emissions of Gases and Aerosols from Nature (MEGAN) version 2.1 (Guenther et al., 2012) which has been integrated into Meso-NH.

## 5.2 Chemistry

The general chemistry equations were first described in the seminal works of Suhre et al. (1995, 1998). The chemistry part of Meso-NH was fully outlined by Tulet et al. (2003) and later completed with the aqueous phase by Leriche et al. (2013). To resolve the coupled differential chemistry equations, several chemical solvers are available such as the QSSA(Quasi-Steady-State Approximation, Hesstvedt et al., 1978) and Rosenbrock families (Sandu et al., 1997) of solvers. QSSA solvers are used for gaseous chemistry simulations whereas the Rosenbrock solvers' are more adapted to address the increase in the system stiffness for cloud chemistry simulations. Photolysis rate coefficients are computed using the TUV (Tropospheric Ultraviolet and Visible radiation) model version 5.0 (Madronich and Flocke, 1999), which can be used on-line or off-line. In order to limit the computational time in 3D simulations, photolysis rates are computed at the first time step for a discrete number of solar zenith angles and altitudes, using ozone and aerosol climatologies and for clear-sky conditions. The choice of ozone and



aerosol climatologies is flexible. Cloud correction of tabulated clear-sky values follows Chang et al. (1987); Madronich and Flocke (1999). In 0D or 1D, the TUV model is used on-line and takes explicitly into account the prognostic ozone and aerosol distributions.

### 5.2.1 Gas-phase chemistry

Several chemical mechanisms are available in Meso-NH (Table 3). The RACM (Regional Atmospheric Chemistry Mechanism, Stockwell et al., 1997) and CACM (Caltech Atmospheric Mechanism, Griffin et al., 2002) mechanisms are largely used in 3D atmospheric chemistry 3D models. The latter is particularly appropriate for the production of semi-volatile precursors of secondary organic aerosols (SOA). Two reduced versions were developed for Meso-NH based on these baseline reaction mechanisms: ReLACS (Regional Lumped Atmospheric Chemical Scheme, Crassier et al., 2000) and ReLACS2 (Regional
Lumped Atmospheric Chemical Scheme version 2, Tulet et al., 2006), respectively .

### 5.2.2 Aerosol module

The different components of the aerosol module ORILAM (ORganic INorganic Lognormal Aerosols Model) are described in Tulet et al. (2005). Only a brief summary of the most important features is given here. A lognormal size distribution function is applied to represent the Aitken, accumulation and coarse modes. The prognostic evolution of the aerosol size
distribution considers three moments for each mode (the zeroth, third, and sixth) to compute the evolution of the total number, number median diameter, and geometric standard deviation. Desert dust and sea salt aerosols are described by three and five lognormal modes, respectively, with a prescribed chemical composition. The size distribution and the chemical composition of anthropogenic aerosols are defined using two lognormal functions for the Aitken and accumulation modes. For these aerosols the chemical mixing is internal and, for each mode, the model computes the evolution of the primary species (black carbon and
primary organic carbon), three inorganic ions ($NO_3^-$, $SO_4^{2-}$, $NH_4^+$), the condensed water, and the 10 SOA classes.

The most important process for the formation of SOA is the homogeneous nucleation in the sulfuric acid-water system. It is based on the Kulmala et al. (1998) parameterization, consistent with the classical theory of binary homogeneous nucleation (Wilemski, 1984), and integrates the hydration effect. The newly formed particles are added to the Aitken mode of anthropogenic particles. The aerosol size distribution evolves via collision between particles, leading to a coagulation process. Both
intramodal and intermodal coagulations are taken into account. Changes in the log-normal distribution are calculated based on Whitby et al. (1991) but modified to allow a particle resulting from two particles colliding within the Aitken mode to be assigned to the accumulation mode. Anthropogenic aerosols are fully coupled with the gas phase chemistry allowing subsequent interactions with gaseous source precursors. The ORILAM scheme assumes that the aerosols are old enough to have a short liquid film at the surface, which favors the absorption process. An inorganic chemistry system calculates the chemical composi-
tion of sulfate-nitrate-water-ammonium aerosols based on equilibrium thermodynamics. Several solvers are implemented such as ARES (Binkowski and Shankar, 1995), ISORROPIA (Nenes et al., 1998) and EQSAM (Metzger et al., 2002). For organics, ORILAM uses the MPMPO scheme (Griffin et al., 2003; Dawson and Griffin, 2016) coupled with the CACM or ReLACS2 chemical schemes (Tulet et al., 2006).





| Mechanism | Number of total prognostic species | Gaz | Aerosol | Aqueous | Number of reactions |
|---|---|---|---|---|---|
| RACM | 105 | 73 | 32 | 0 | 240 |
| ReLACS | 69 | 40 | 32 | 0 | 128 |
| CACM | 241 | 189 | 52 | 0 | 349 |
| ReLACS2 | 134 | 82 | 52 | 0 | 343 |
| ReLACS-AQ | 123(142) | 41 | 32 | 50 | 272 |
| ReLACS3 | 214(245) | 88 | 52 | 74 | 581 |

**Table 3.** Chemical mechanisms available in Meso-NH with the number of total prognostic species, the decomposition between gas, aerosols and aqueous species, and the number of reactions. For ReLACS-AQ and ReLACS3, the numbers in parenthesis include the precipitating ice mixing ratios for mixed phase clouds.

## 5.3 Impact of clouds

A detailed approach to wet deposition is implemented in Meso-NH taking full advantage of access to microphysical tendencies and microphysical reservoirs. For gases, the sink via wet deposition includes an explicit computation by the cloud chemistry module for the resolved clouds whatever the microphysical scheme used including mixed-phase processes (Leriche et al., 2013)
and mass-flux parameterization for sub-grid scale convective clouds (Mari et al., 2000). For aerosols, wet deposition is considered via impaction scavenging and aerosol activation. For example, aqueous-phase chemistry is crucial to the production of SOA. Leriche et al. (2013) provide a comprehensive description of the aqueous-phase chemistry module. The pH is diagnosed by solving the electroneutrality equation. Two chemical mechanisms were developed to account for the aqueous-phase reactions based on the ReLACS and ReLACS2 mechanisms. ReLACS-AQ incorporates the aqueous chemistry based upon Tost
et al. (2007) and CAPRAM2.4 (Chemical Aqueous Phase RAdical Mechanism version 2.4, Ervens et al., 2003). ReLACS3 (Regional Lumped Atmospheric Chemical Scheme version 3, Berger, 2014) integrates organic chemistry from CLEPS (Cloud Explicit Physico-cehmical Scheme, Mouchel-Vallon et al., 2017). Below-cloud impaction scavenging is described in details in Tulet et al. (2010). The in-cloud aerosol mass transfer into rain droplets via autoconversion and accretion processes has been incorporated as described by Pinty and Jabouille (1998). Two options are available for aerosol activation in warm clouds. In the
first method, the total particle number of the accumulation modes calculated by ORILAM are transfered into the LIMA CCN classes (sea salt, sulfates, and hydrophilic organic matter and black carbon) according to their chemical composition. Then the CCN activation follows the activation scheme of LIMA. The second method takes full advantage of the chemical composition and the size distribution of each mode to compute the Raoult and Kelvin terms of the Köhler theory (Köhler, 1936). The CCN activation scheme is based on Abdul-Razzak and Ghan (2004). In this method, ORILAM computes the number of dissociative
ions, soluble fraction of each aerosol compound, organic surfactants and lognormal parameters for each mode. For ice nucleation, the Aitken and accumulation modes of dust particles and hydrophobic organic matter and black carbon are placed in the corresponding IFN classes of LIMA. The nucleation scheme follows Phillips et al. (2008).





## 6 Diagnostics

One strength of Meso-NH as a research model is that it offers a rich palette of diagnostics and statistics to sample simulations, facilitate comparisons to observational data of experimental field campaigns, or scrutinize the source and sink terms of prognostic fields. Numerous observation operators have also been developed to compare the model output directly to satellite,

radar, LIDAR, and Global Positioning System (GPS) observations and to constitute a first step toward the assimilation of these types of observational data into operational NWP models such as AROME. A few examples of the diagnostic capabilities of Meso-NH are given below.

### 6.1 Diagnostics, spectra, and budgets

Sharing Meso-NH with the research community leaves the code with a large set of diagnostic fields to be computed in post-

processing. The energy spectrum can be derived from the wind, temperature or humidity fields according to Ricard et al. (2013) (e.g., the kinetic energy spectra plotted in Fig. 4). During runtime, a module can provide the fully closed budget of all the prognostic fields, which can be computed over Cartesian boxes or masks, allowing the calculation of conditional statistics, e.g., updrafts, clouds or intense surface precipitation.

### 6.2 Passive tracers and dispersion modeling

Meso-NH delivers the necessary tools to study the dispersion of passive tracers using the Eulerian and Lagrangian frameworks. Eulerian passive tracers are easily addressed giving the characteristics of a release. An original method for tracking coherent Lagrangian airmasses has been introduced by Gheusi and Stein (2002) based on three Eulerian passive tracers initialized with the coordinates of each grid cell. Each Lagrangian air parcel is identified by its initial position so that its physical history can be retrieved. Resolved and subgrid (turbulence, convection) transports are taken into account, enabling the technique to study

forward and backward motions. A few illustrations of the method capabilities can be found in Ducrocq et al. (2002), Colette et al. (2006), Chaboureau et al. (2011), Duffourg et al. (2016), and Vérèmes et al. (2016).

Meso-NH is used for environmental emergencies because Météo-France, as a civil security organization, needs to predict contaminated areas subsequent to accidental releases, from the close-to-source (near 2 km) area to the regional scale. Meso-NH, running at 2-km horizontal resolution, is combined to a Lagrangian stochastic dispersion model in an integrated modeling

system to be able to simulate and track accidental airborne pollutants anywhere on Earth (Lac et al., 2008). Figure 7 illustrates the dispersion of a smoke cloud resulting from a lava flow on the southeast slopes of the Piton de la Fournaise volcano on 18 May 2015 over Reunion Island. The plume rounded the volcano from the south before being taken into the stream of the trade winds.

Meso-NH has also been used to simulate atmospheric $CO_2$ concentrations under various mesoscale flow conditions and surface

area to improve our understanding of the terrestrial carbon budget (Sarrat et al., 2007a, 2009b, a; Lac et al., 2013). Forward simulations have provided support for regional inversions with networks of $CO_2$ observations to retrieve fossil fuel $CO_2$





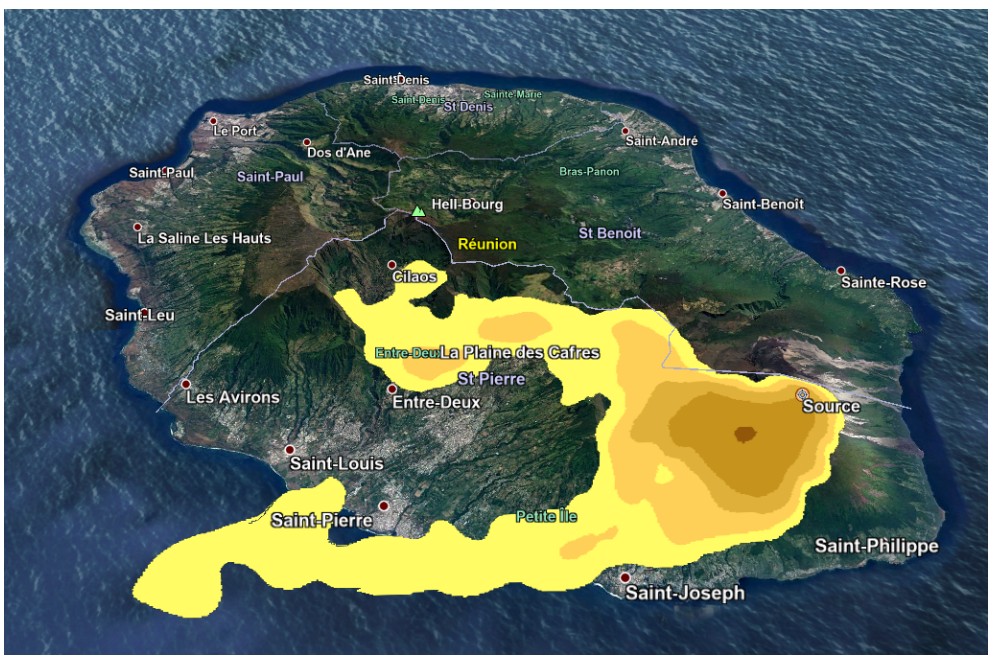

**Figure 7.** Atmospheric transfer coefficient (in s m$^{-3}$) normalizing the concentration with the emission flow rate during the 6 h following 00:00 UTC on 18 May 2015 (isolines with logarithmic intervals from $10^{-14}$ s m$^{-3}$ to $10^{-9}$ s m$^{-3}$).

sources and sinks (Lauvaux et al., 2008, 2009b, a and Staufer et al., 2016 using 1-year-long kilometric simulations over the Paris region).

## 6.3    Aircraft, balloons, and profilers

In order to compare the model outputs to airborne measurements, it is possible to simulate the travel of a balloon or an aircraft during the run in any nested model, e.g., while considering the balloon's density (an iso-density balloon), particular volume (a constant volume balloon), and ascent speed (radio-sounding). All the prognostic fields are recorded along the trajectory of the balloon or aircraft. Temporal series over single points or averaged over a Cartesian area can also be recorded to compare to profilers or station measurements.

## 6.4    LES diagnostics and conditional sampling

LESs allow the separation of resolved and subgrid parts of a field, to characterize its fine-scale variability in order to develop parameterizations or to identify coherent structures. Diagnostics can be included in standard output files including time series and averaged profiles of mean variables, (co-)variances, resolved and subgrid fluxes, and PDFs of dynamical and thermodynamical fields within all or a part of the simulation domain. A conditional sampling based on the emission of a passive tracer at the surface according to Couvreux et al. (2010) is proposed to characterize coherent structures in LESs of cloud-free and



cloudy boundary layers. This allows the identification of convective updrafts from the surface to the top of the boundary layer and the characterization of plumes, entrainment and detrainment rates, variances, and fluxes. This method has been used by Rio et al. (2010) to evaluate the EDMF parameterization and by Perraud et al. (2011) and Jam et al. (2013) to develop the PDF of the saturation deficit in LES convective BLs. Honnert et al. (2016) adapted conditional sampling to detect the subgrid

component of thermals at a given spatial resolution.

### 6.5   Coarse-graining techniques

Coarse-graining techniques calculate the average and standard deviation of any model field over a set of user-defined blocks. Such techniques are useful when developing a subgrid parameterization and are commonly applied to a set of two simulations that differ only in their resolution. The high-resolution simulation provides the average fields on a coarse grid that should

be obtained by the low-resolution simulation run with the subgrid parameterization to be tested. The operator is a parallel algorithm that can easily be employed over large grids. The operator can also calculate a moving average over a user-defined block. Both the grid-scale and the subgrid-scale of any field can therefore be estimated (Dauhut et al., 2016).

### 6.6   3D clustering

A clustering operator is available to identify any object or coherent structure and to characterize them in terms of their geomet-

rical, thermodynamical, and dynamical properties. This technique was developed by Dauhut et al. (2016) to identify the few updrafts of Hector the Convector in the Northern Territories of Australia from among the more than 16,000 updrafts that hydrate the stratosphere. Updrafts were defined as three-dimensional objects made of connected grid points for which the vertical velocity exceeded an arbitrary threshold. Two grid points sharing a common face either in the horizontal or vertical direction were considered connected, while diagonal connections were considered only in the vertical direction. This technique has also

been used for the attribution of dust emission, defined as surface objects, to wind regimes over the Sahara (Chaboureau et al., 2016).

### 6.7   Observation operators

Synthetic brightness temperatures (BTs) for satellite infrared or microwave nadir scanning radiometers can be computed offline using the Radiative Transfer for Tiros Operational Vertical Sounder (RTTOV) code version 11.3 (Saunders et al., 2013). RTTOV

uses the atmospheric profile of temperature, water vapor, cloud and precipitating hydrometeors, and surface properties predicted by the model. RTTOV is a powerful tool for verifying the realism of simulations by comparing observed and synthetic BTs (e.g., Chaboureau et al., 2008). An example is given in Fig. 8a for HyMeX IOP16. The satellite operator has been used to develop the representation of clouds in Meso-NH. The ice-to-snow autoconversion threshold in the ICE3 scheme has been tuned once for midlatitude storms (Chaboureau et al., 2002) and once for the tropical atmosphere by introducing a dependence

on the temperature (Chaboureau and Pinty, 2006). The deep convective variance introduced into the subgrid cloud scheme was assessed against satellite observations (Chaboureau and Bechtold, 2005).



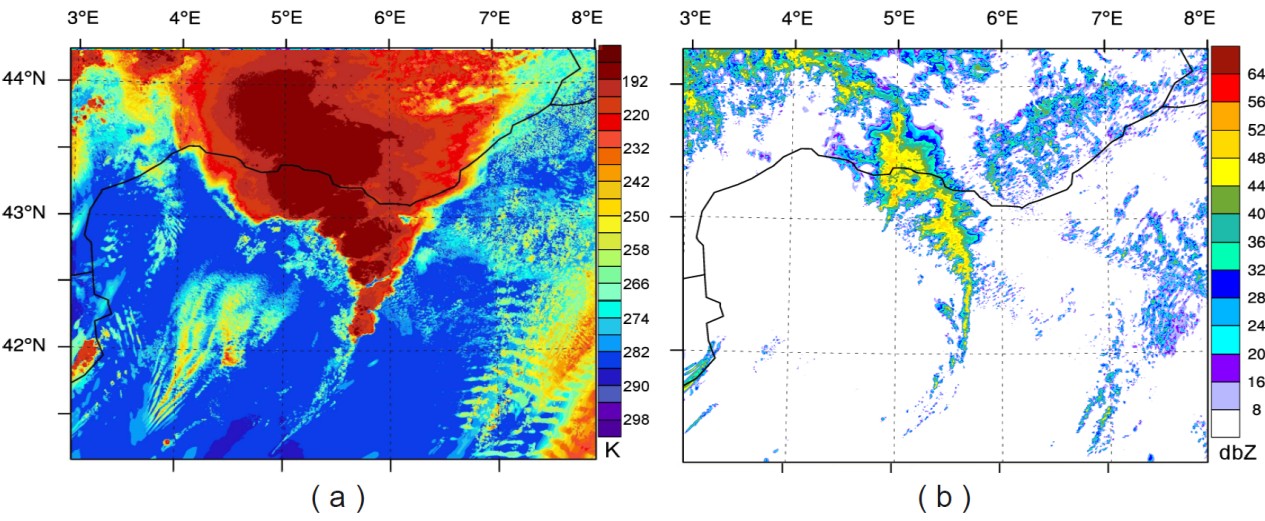

**Figure 8.** (a) Brightness temperature (in K) and (b) the 850 hPa radar reflectivity (in dBz) simulated by Meso-NH at a 150 m horizontal resolution on 26 October 2012 at 11:00 UTC (HyMeX IOP16).

A forward observation operator for dual-polarization radars has been developed in the model, suitable not only for a variety of operational weather radars (S-, C-, and X-bands, Augros et al., 2016) but also for airborne cloud radars at W-band. The forward operator is consistent with the microphysical schemes ICE3 and LIMA. All dual-polarization variables measured by the radars are simulated: horizontal reflectivity $Z_{hh}$, differential reflectivity $Z_{dr}$, differential propagation phase shift $\phi_{dp}$, the
copolar correlation coefficient $\rho_{dp}$, specific differential phase shift $K_{dp}$, specific attenuation $A_{hh}$, and differential attenuation $A_{dp}$, as well as the back-scattering differential phase $\delta_{hv}$. Extensive comparisons between the observed and simulated radar variables were performed during the first observing period of the HyMeX experiment (Ducrocq et al., 2014) using ground-based dual-polarization radars (Augros et al., 2016) (Fig. 8b) and the airborne cloud radar RASTA (RAdar SysTem Airborne, Duffourg et al., 2016). The radar operator is a very useful tool to evaluate the 3D hydrometeor characteristics in the model and
to test the microphysical parameterizations.

A LIDAR emulator computes the attenuated backscattered signal corrected for geometric effects and a calibration constant (Chaboureau et al., 2011). The extinction and backscatter coefficients caused by air molecules, aerosols, and cloud particles are calculated online. The optical properties of the cloud particles and aerosols are integrated over their size distribution. For the two-moment microphysical schemes, the integration is performed using an accurate quadrature formula. For the single-
moment microphysical schemes, it is computed taking an effective radius representative of the distribution. The extinction and backscatter efficiencies of the cloud particles and aerosols are computed using a Mie code depending on their refractive index. The emulator is suitable for any nadir- or zenith-pointing LIDAR system, as shown in the assessment of a simulation of the long-range transport of dust (Chaboureau et al., 2011).



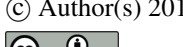

| Processes simulated by Meso-NH | Applications | Section or example of references |
| --- | --- | --- |
| Surface | Urban meteorology and climate studies | 7.2.1 |
| | Ocean coupling, hurricanes | 7.1 |
| | Ecosystem studies | Sarrat et al. (2007b) |
| Clouds | Weather process studies | Duffourg and Ducrocq (2011) |
| | Climate process studies | Khodayar et al. (2016) |
| | Hydrology | Vincendon et al. (2009); Hally et al. (2015) |
| | Hurricanes | Coronel et al. (2015) |
| | Fog | Mazoyer et al. (2017) |
| Turbulence | Weather process studies | Cuxart and Jiménez (2007) |
| | Optical turbulence for astronomy | 8.3 |
| Wildland fire | Air quality, weather impacts | 7.3 |
| Passive dispersion | Air quality, accidental release forecasting | 6.2 |
| Dust and sea salt | Air quality, climate impacts | Tulet et al. (2010); Bègue et al. (2012) |
| Wind-induced snow transport | Snow studies, avalanche danger, hydrology | 7.2.2 |
| Chemistry/aerosols | Atmospheric chemistry research, air quality | 7.5 |
| | Volcanoes | 7.5 |
| Electricity | Weather and process studies | 7.5 |
| | Atmospheric chemistry research | Barthe et al. (2007) |
| | Hurricanes | Barthe et al. (2016b) |

**Table 4.** Main capabilities and applications of Meso-NH version 5.4.

# 7 Innovative couplings and large grid applications

Table 4 lists the main process capabilities and applications of Meso-NH. This section highlights some recent developments in the coupling of Meso-NH with other models together with some applications of Meso-NH over large grids. Several of these developments are innovative, such as wind-induced snow transport and fire propagation, while others are used in very different contexts and resolutions that make them completely original.

## 7.1 Oceans and waves

The coupling interface developed in SURFEX by Voldoire et al. (2017) allows Meso-NH to be coupled to any ocean or wave model that includes OASIS3-MCT coupler (Valcke et al., 2015) code instructions, for example, the NEMO (Madec, 2008), MARS3D (Lazure and Dumas, 2008), and SYMPHONIE (Marsaleix et al., 2008) ocean models and the WW3 wave model (Tolman et al., 2009). The OASIS3-MCT coupler is a library using MPI, which allows the coupling of any model with a minimal amount of changes. The coupler exchanges and interpolates fields between Meso-NH and the ocean and/or wave model. Specifically, the air–sea fluxes computed by SURFEX on the Meso-NH grid, and other Meso-NH atmospheric



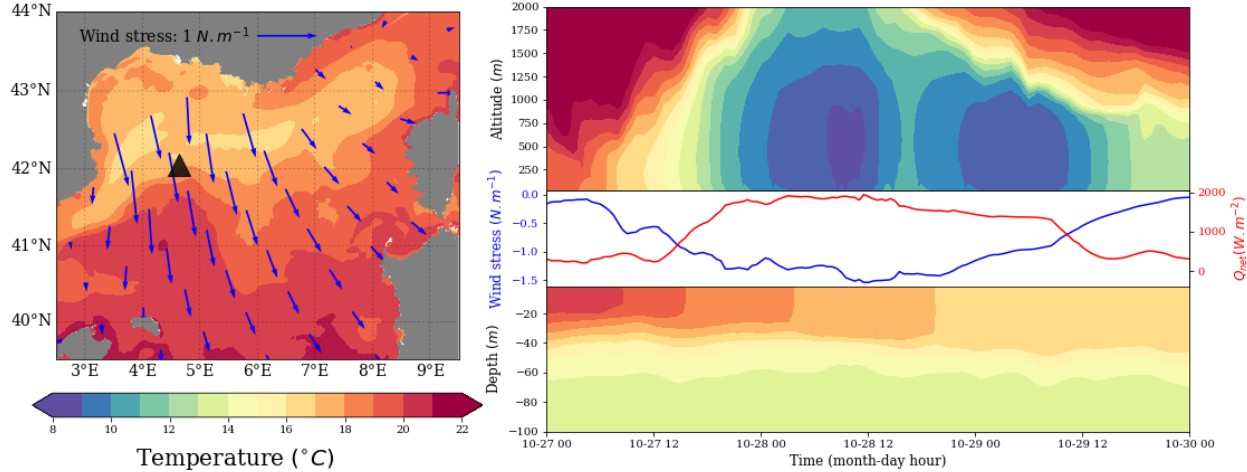

**Figure 9.** Results from the coupling between Meso-NH and SYMPHONIE. Left panel: sea surface temperature (color, °C) and 10 m wind (vector, $\mathrm{m\,s^{-1}}$) averaged over 27–30 October 2012. The triangle symbol shows the location of the LION buoy. Right panel: time evolution of the (top) air temperature (°C), (middle) wind stress (blue, $\mathrm{N\,m^{-1}}$) and sensible heat flux (red, $\mathrm{W\,m^{-2}}$), and (bottom) water temperature (°C) at the LION buoy.

variables needed to drive the ocean and/or wave models, are interpolated to ocean/wave model grids and sent to them by the OASIS3-MCT coupler. Conversely, the sea surface temperature and currents computed by the ocean model, and the wave parameters computed by the wave model, are interpolated to the Meso-NH grid and sent to SURFEX by the OASIS3-MCT coupler. Voldoire et al. (2017) demonstrated various applications of Meso-NH coupled with either the NEMO, SYMPHONIE,

MARS3D, or WW3 models, which enabled the study of ocean–wave–atmosphere processes at various scales and their impacts on the atmosphere in response to a sea surface temperature front over the Iroise Sea, a tropical cyclone over the Indian Ocean, and severe Mediterranean weather events.

An example of ocean coupling is shown here with SYMPHONIE over the Mediterranean Sea during the Mistral and Tramontane event of 27–30 October 2012 (Fig. 9). The Meso-NH–SYMPHONIE coupled system shows the southward advection

of cold air that led to a large decrease in the air temperature by more than 10°C in 36 h at the LION buoy location. The sea temperature in the first 20 m also significantly decreased by more than 4°C in 36 h due to the vertical turbulent heat flux and the complex interaction between the vertical (turbulent heat flux and Ekman pumping) and horizontal (fine-scale structure displacement and frontal dynamics) processes. The coupling allows the representation of these fine-scale and complex ocean dynamics and responses and led here to a decrease [increase] in the oceanic [atmospheric] boundary-layer temperature, there-

fore reducing the oceanic surface-layer instability and inhibiting the atmospheric turbulent heat flux (and turbulent moisture flux, not shown). This illustrates that a 3D air–sea coupling is essential to sufficiently represent the heat (and moisture) budget in atmospheric and oceanic boundary layers during strong wind events. Note that it is necessary to study the process of open ocean convection in the northwestern Mediterranean Sea.





### 7.2 Continental surfaces

### 7.2.1 Urban studies

The Meso-NH-SURFEX coupling offers a wide range of applications over continental surfaces. Urban meteorology constitutes one such application because cities modify the local meteorology, creating their own microclimate, such as the Urban Heat

Island (UHI). Lemonsu and Masson (2002) presented the world's first UHI mesoscale simulation coupled with an urban model (TEB), which was able to numerically reproduce an UHI of 8 K for the agglomeration of Paris. Since then, Meso-NH has been used to analyze various urban climate processes, e.g., air pollution (Sarrat et al., 2006), the vertical structure of the boundary layer of coastal cities (Lemonsu et al., 2006b, a; Pigeon et al., 2007), and urban breeze (Hidalgo et al., 2008, 2010). More recently, the ability to perform hectometric resolution simulations opened a new field of urban climate research: the study

and multi-scale evaluation of adaptation strategies of cities to climate change (Lemonsu et al., 2013). De Munck et al. (2013) showed that air-conditioning systems would increase the night-time air temperature by 1-2 K during a heat-wave episode. This air temperature increase is larger during the night than during the day, which may be counterintuitive; however, this is due to the vertical structure of the boundary layer. Green roofs and other vegetation strategies, as well as agglomeration-wide urban planning strategies have also been evaluated (Masson et al., 2013b; Daniel et al., 2016). Year-long hectometric-scale

simulations are now performed in order to evaluate the urban microclimate and its impacts (such as the energy consumption of buildings, Schoetter et al., 2017). This creates the possibility of building new methodologies of regional climate downscaling down to urban and intra-urban scales. Figure 10 shows a comparison between the near-surface temperature simulated by Meso-NH-TEB and those observed during the CAPITOUL intensive observation campaign (Masson et al., 2008) during the spring of 2004 for two weather types. Despite a positive bias in the absolute values of the air temperature, Meso-NH-TEB captures

the sensitivity of the air temperature to the building density well.

### 7.2.2 Wind-induced snow transport

As presented in Section 5.1, Meso-NH is coupled with SURFEX to model the emission of natural aerosols such as desert dust and sea salt. In the same way, another coupling concerns the wind-induced snow transport via the detailed snowpack model Crocus (Brun et al., 1992) of SURFEX. Meso-NH–Crocus simulates snow transport via saltation and turbulent suspension

and includes the sublimation of suspended snow particles (Vionnet et al., 2014). In the atmosphere, blown snow particles are represented by a two-moment scheme to capture the spatial and temporal evolution of the particle size distribution. At the surface, the model computes the mass flux in saltation as a function of the snow-surface properties simulated by Crocus and the near-surface meteorological conditions. Finally, the model simulates snow erosion and deposition including the contributions of saltation, turbulent suspension, and snowfall simulated by Meso-NH. Meso-NH–Crocus has been used down to a grid spacing

of 50 m to simulate snow redistribution during blowing snow events in alpine terrains. In particular, it has been used to quantify the mass loss due to blowing snow sublimation (Vionnet et al., 2014) and to study the spatial variability of snow accumulation (Vionnet et al., 2017).





**Figure 10.** Sensitivity of the nocturnal (1 to 5 Local Solar Time) near surface air temperature on urbanization for a domain covering the agglomeration of Toulouse (France) at a horizontal resolution of 250 m. (a) and (c): Difference in the 2 m air temperature between a simulation taking the urbanization into account via TEB and a simulation without urbanization for all days during March, April, and May 2004 classed into (a) relatively windy and cloudy days and (c) calm and sunny days via the clusterization of Hidalgo et al. (2014). (b) and (d): Comparison between the values of the air temperature 6 m above the ground simulated by Meso-NH-TEB and observed during the CAPITOUL campaign for the same two weather types. The locations of the stations are displayed in the left column.



### 7.3 Fire propagation

Numerous observational studies (Clements et al., 2006; Santoni et al., 2006) have shown that strong interactions exist between wildfires and the atmosphere at different scales (turbulent mixing in the front, large eddies near the front, fire-induced winds, and pyrocumulus clouds). Wildland fire is a multiscale process, from the flame reaction zone on sub-meter scales to the synoptic scale of hundreds of kilometers. The numerical coupling between a fire model and an atmospheric one is a good way to understand the mechanisms driving a fire spread and has research implications for operational fire spread models. Numerical fire–atmosphere coupling has already undergone numerous developments, starting from the static fire simulations of Heilman and Fast (1992) to more recent studies where a simplified fire spread model is coupled with an atmospheric mesoscale model (Mandel et al., 2011) running at a regional scale. The objective here was to develop this type of two-way interactive coupling with a more physical fire spread model and to run at the scale of the fire front, i.e., with LESs (Filippi et al., 2009). Meso-NH has accordingly been coupled with ForeFire (Balbi et al., 2007), a physical fire spread model taking into account wind and slope effects. In ForeFire, the fire front acts as a tilted radiant panel that heats the vegetation in front of it, vaporizing the water content before entering pyrolysis. Wind and slope effects are explicitly taken into account by calculating the flame tilt angle using a vector method. The rate-of-spread (RoS) for every portion of the front is then used by a front-tracking method to simulate the fire perimeter. At each time step of the atmospheric model, Meso-NH forces the fire behavior via the surface wind field, whereas the fire forces the atmospheric simulation via the surface heat and vapor fluxes through SURFEX. The coupling involves extreme values for the atmospheric model, such as surface temperatures on the order of $1000\,\mathrm{K}$ corresponding to upward radiative fluxes 100 times larger than normal, or upward sensible fluxes 100 times larger than normal (up to $100\,\mathrm{kW\,m^{-2}}$ at resolutions of 50 m). The coupled Meso-NH–ForeFire system has been validated with idealized simulations showing strong interactions between the topography and the fire front induced wind (Filippi et al., 2009, 2011) in the experimental burn of FireFlux (Filippi et al., 2013) and in real cases located in the Mediterranean region (Filippi et al., 2011). Strada et al. (2012) explored the air quality in addition to the dynamics downwind of a burning area, including the atmospheric online gaseous chemistry, with Meso-NH.

Another challenge has been to run Meso-NH–ForeFire on the Aullene wildland fire in Corsica, which occurred on 23 July 2009 and burned 2000 ha during the first afternoon. The simulation included four nested domains from the regional scale (2400 m horizontal resolution) to the fire scale (50 m resolution) in a two-way configuration; the combined model received the second Bull-Fourier price in 2014 for its run on massively parallel computers. The burnt area was reproduced with a good degree of realism at the local scale (Fig. 11a) and at the regional scale, where the simulated fire plume compares well with the MODIS satellite image (Fig. 11b).

In 2007, the coupled system was extended to simulate the progression of the lava flow and the smoking plume during the eruption of the Piton de la Fournaise volcano on Reunion Island (Durand, 2016).

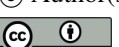



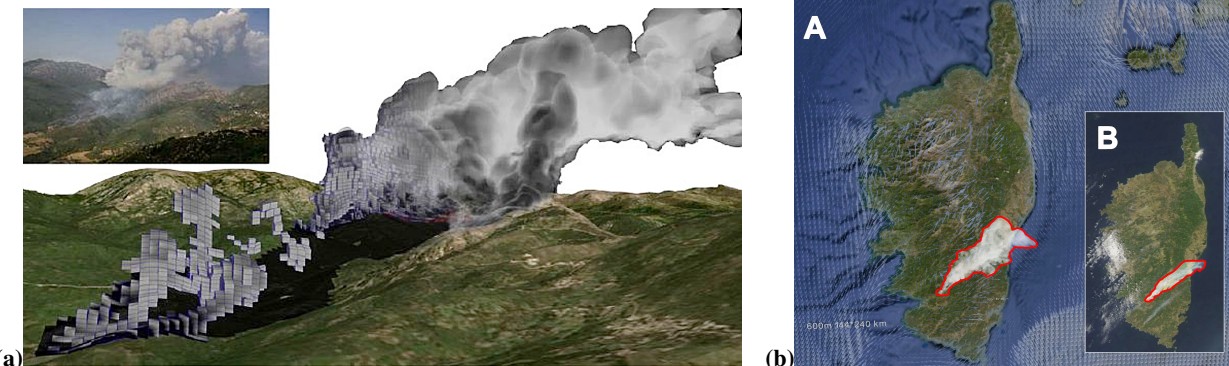

**Figure 11.** Simulated smoke tracer on 23 July 2009 (a) in the 50 m resolution domain compared to the plume's photograph (at the top left) and (b) in the 600 m resolution domain highlighted in red (A) at 15:00 UTC compared to the MODIS image (B) of Corsica at 14:50 UTC.

## 7.4 Electricity

Explicit simulations of electrified clouds with CELLS in Meso-NH have first investigated idealized convective cases to understand the physical processes driving the elctrical properties of the clouds and to test their sensitivity. They now begin to study the electrification of real meteorological precipitating cloud systems, including comparison with electrical observations.
CELLS has successfully reproduced the electrical activity of several idealized storms (Barthe and Pinty, 2007b; Tsenova et al., 2017), an idealized tropical cyclone-like vortex (Barthe et al., 2016a) and the 21 July 1998 EULINOX storm (Barthe et al., 2012a). The modeling of the production of nitrogen oxides by lightning flashes was realized and illustrated for the 10 July 1996 STERAO storm (Barthe et al., 2007; Barth et al., 2007a). Pinty et al. (2013) realistically simulated the electrical aspects of a heavy precipitation event over the Cevennes area in the South of France in the HyMeX experiment. More recently, a few
cases have been under investigation over Corsica taking advantage of the lightning-observing network SAETTA. This network, made up of 12 Lightning Mapping Array (LMA) stations, has been deployed in Corsica to monitor the 3D lightning activity within a range of approximately 350 km from the center of Corsica. The SAETTA dataset can therefore be used to assess in details the functioning of CELLS for multiple events.

As an example, Meso-NH was able to reproduce the electrical properties of a local convective development in the afternoon
of 25 July 2014 over northern Corsica. For the entire event, the SAETTA estimate was $\sim$ 1050 flashes while the model reproduced $\sim$ 850 flashes. In Fig. 12, the sequence of the VHF records of the SAETTA profiles shows the two-level propagation of the flashes. In the first window highlighted by the bold rectangle, more flashes are observed at high level $\sim$ 9 km (red color) meaning an excess of positive charges. Conversely, the second window shows that later, the polarity of the charge doublet reverses so that the charges at the top are negative. This charge reversal was well captured by CELLS in Meso-NH when
simulating the case at a 1 km resolution. On the left, a "direct" electrical cell appears, while one hour later a negative charge density overhangs the positive pocket of charges on a shifted cross-section.



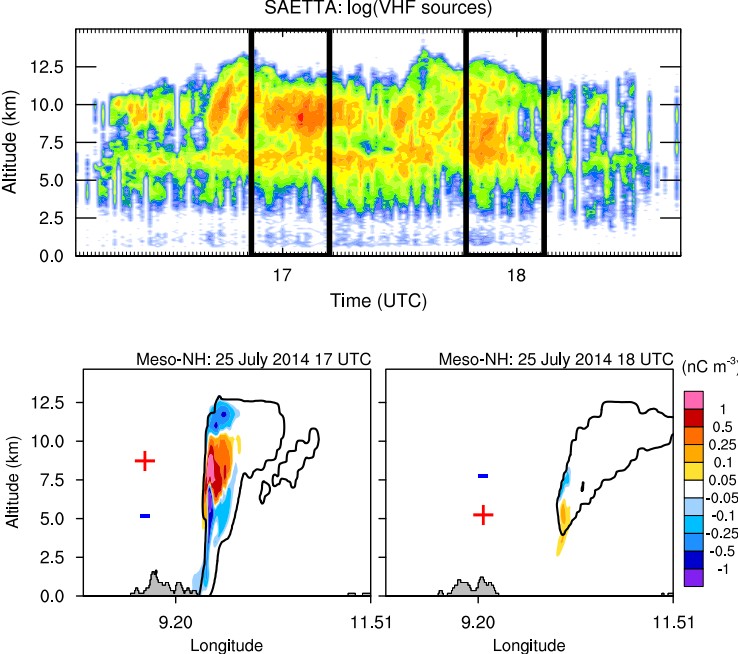

**Figure 12.** Comparison between the SAETTA data and the Meso-NH simulation for the case of the 25 July 2014 event over Corsica. Time series of the vertical profiles of the SAETTA VHF sources (top) and MesoNH cross-sections of the total charge density (nC m$^{-3}$; colors) corresponding to the windows of the SAETTA profiles with a direct cell (left) of "normal" polarity and an indirect cell (right) of "reverse" polarity. The cloudy area is shown with a black isoline.

## 7.5 Chemistry and aerosols

Meso-NH is applied in a wide range of research on air quality and climate process studies as it handles gaseous and aqueous chemistry and aerosols. Figure 13 shows a 2D view of the SOA mass concentration of the class 6 SOA upstream of the Puy de Dôme mountain, at the summit and downstream, as well as the relative contribution of the 10 SOA classes to the total mass. These results were obtained with a 2D idealized simulation in a plane parallel to the main wind direction (Berger et al., 2016b). The isoprene mixing ratio was initialized from measurements performed at the Puy de Dôme station for a particular orographic cloud observed in July 2011. However, for this event, the isoprene mixing ratio was very weak and a sensitivity test was performed multiplying the isoprene mixing ratio by 20. For both cases, the production of the class 6 SOA is observed with the relative contribution of this class increasing downstream of the mountain. Class 6 SOA is produced from oxalic and pyruvic acids, which are produced inside cloud droplets from the oxygenated soluble isoprene oxidation products. Multiplying the initial mixing ratio of isoprene by 20 leads to only a doubling of the mass concentration of class 6 SOA downstream of the



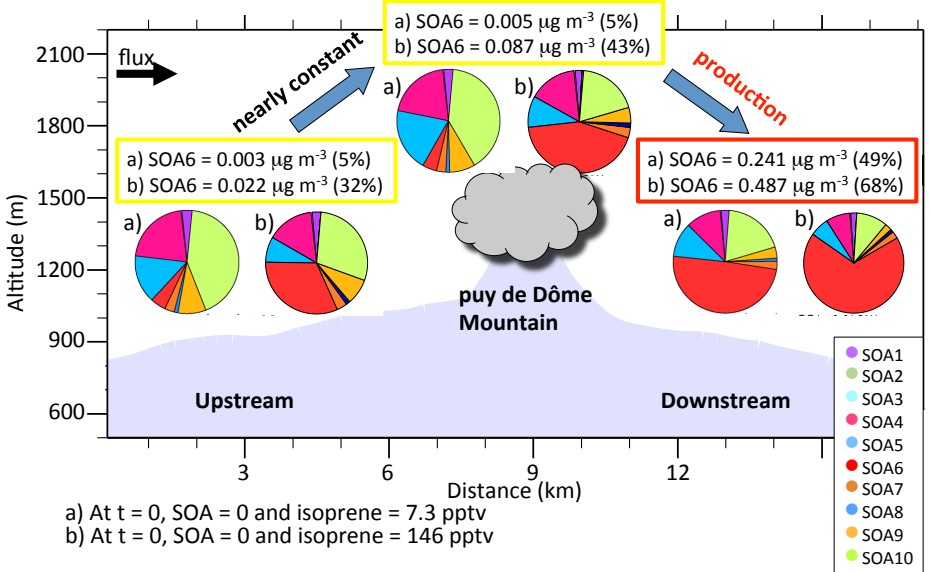

**Figure 13.** Mass concentration of the class 6 secondary organic aerosol (SOA6) upstream, at the summit and downstream of the Puy de Dôme mountain (rectangular box). The contribution of the 10 classes of secondary organic aerosols to the total mass is represented by a pie-chart for (a) an isoprene initial mixing ratio of 7.3 pptv and (b) an isoprene initial mixing ratio of 146 pptv (multiplied by 20).

mountain. This is likely because the gaseous chemistry upstream leads to the significant production of oxalic and pyruvic acids as indicated by the mass concentration of the class 6 of SOA upstream of the mountain where the isoprene mixing ratio is the highest.

Volcanoes are one of the most important natural sources of air pollution. It is crucial for air quality, aviation hazard forecast-
5  ing, and climate studies to have a good knowledge of their atmospheric chemistry, physical, and radiative effects. Figure 14 shows a 3D view of the $SO_2$ concentration from the Etna (Italy) volcanic plume modeled by Meso-NH at 2 km horizontal resolution on 15 June 2016 at 14:00 UTC during the 2016 STRAP campaign (http://osur.univ-reunion.fr/recherche/strap/). The $SO_2$ concentration observed by the SAFIRE ATR42 aircraft is superimposed on the figure. The $SO_2$ concentration modeled for the plume is close to the observations. The horizontal gradients of $SO_2$ are well located, and the $SO_2$ concentration decreases
10  into the plume as was observed. This shows that Meso-NH is able to correctly reproduce the transport, dilution and chemical transformation of the volcanic plume. It simulates 911 ppb of $SO_2$ above the vent, and 100 ppb and 20 ppb of $SO_2$ at dis-
tances of 4 km and 120 km from the vent, respectively. The ReLACS2 chemical scheme of Meso-NH transforms the $SO_2$ into



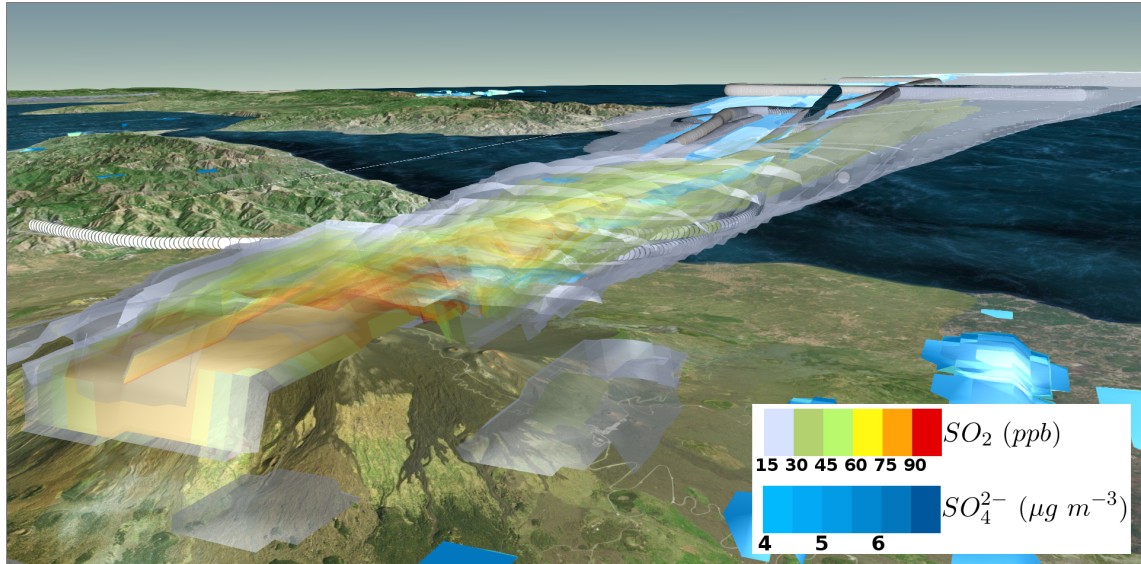

**Figure 14.** The $SO_2$ concentration in the Etna volcanic plume modeled by Meso-NH on 15 June 2016 at 14:00 UTC (in ppb, scale in color), and the $SO_4^{2-}$ concentration ($\mu g\,m^{-3}$, scale in blue gradient) in the aerosol phase. The trajectory of the SAFIRE ATR42 aircraft is superimposed by using circles, and the colors represent the observed $SO_2$ concentration (in ppb, scale in color).

sulfuric acid ($H_2SO_4$). Then, the aerosol scheme ORILAM nucleates and condenses the sulfuric acid into the aerosol aqueous phase ($SO_4^{2-}$ions ). Meso-NH produces the maximum value of the $SO_4^{2-}$ concentration ($9\,\mu g\,m^{-3}$) 114 km from the vent. In addition, close to the surface, the air pollution from the Catania (Italy) region is simulated.

## 7.6   Large computational grid applications

The recent advent of massively parallel computers, using hundreds of thousands of cores, has opened new possibilities. These computers are now sufficient to perform seamless modeling of weather events and to study their scale interactions over large grids (Pantillon et al., 2013; Paoli et al., 2014; Bergot et al., 2015; Dauhut et al., 2015). Pantillon et al. (2013) ran a convection-permitting simulation of Hurricane Helene (2006) and its interaction with a planetary wave using $4\,\mathrm{kcores}$. They used a domain with 412 million points ($3072 \times 1920 \times 70$) that stretched from the eastern Pacific to the western Mediterranean and showed

that the 5-day track of Helene could be correctly forecasted when running the model at high resolution. Paoli et al. (2014) carried out LESs of stably stratified flows and discussed the impact of resolution by increasing the number of points to 8.59 billion points ($2048 \times 2048 \times 2048$) and the number of cores up to $4\,\mathrm{kcores}$ while decreasing the grid spacing down to $2\,\mathrm{m}$. Bergot et al. (2015) performed LESs of radiation fog over an airport area to study the effect of an urban canopy on the fog. Using a domain of 425 million points ($3072 \times 1024 \times 135$), they demonstrated the advantage of using LESs on complex terrains

to better understand fog physics. Dauhut et al. (2015) ran a simulation of Hector the Convector, an Australian multicellular thunderstorm, over a domain of 1.34 billion points ($2560 \times 2048 \times 256$) and a grid mesh of $100\,\mathrm{m}$ (Fig. 15). By contrasting



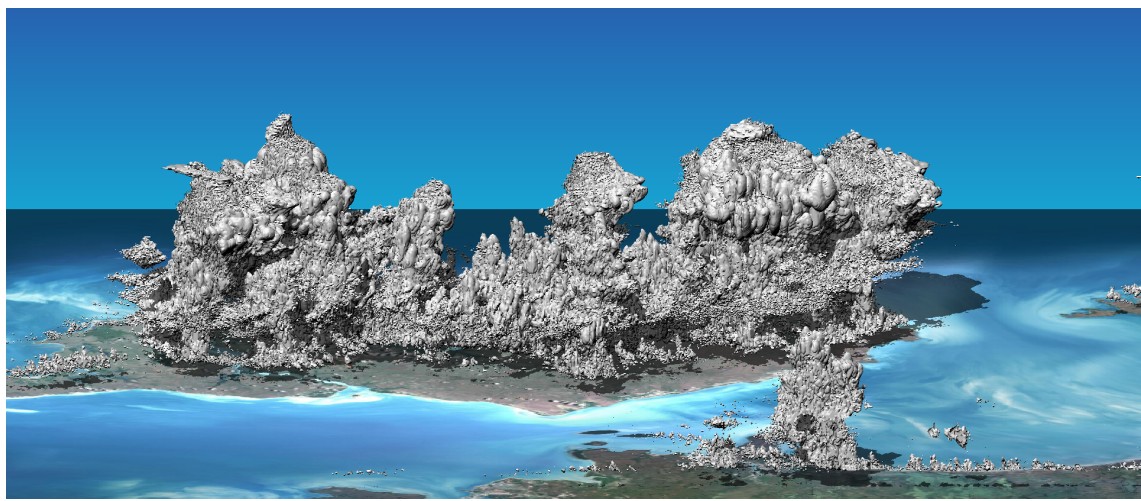

**Figure 15.** Snapshot of a Meso-NH simulation of Hector the Convector taken from the 1-minute cloud envelope animation available at https://youtu.be/xjPumywGaAU.

their so-called giga-LESs with runs performed at coarser resolution, they showed that grid spacing on the order of 100 m is necessary to make a reliable estimate of the convective hydration of the tropical stratosphere by a very deep thunderstorm. These studies all show that such LESs are very useful to better understand the mechanisms involved in the processes described above. Because they provide a consistent description of the atmosphere, they can serve as a virtual field campaign. Therefore,

the use of such LESs will likely significantly increase in the near future.

Since Pantillon et al. (2011), Meso-NH has also been used over very large grids at kilometric resolutions to study clouds and convection. Establishing a better knowledge of cloud microphysics and rain production remains the main use of the model. Beyond the Mediterranean cases previously mentioned, another common study region of Meso-NH is over the tropics where convection is ubiquitous. An example is shown for a dusty outbreak at 12:00 UTC on 12 June 2006 over the northern part of

Africa (Fig. 16). As expected in summer, clouds and precipitation occur mainly along the intertropical convective zone, while dust is present over the Sahara. A mesoscale convective system is located over the Sahel, in the middle of the domain between the intertropical convective zone and the Sahara. Such a propagative system is easily obtained with Meso-NH running as a CRM because the coupling between the synoptic circulation and the convective systems is explicitly represented with such a kilometer-scale grid mesh.

# 8   Model evaluation

Evaluations are essential, necessary activities to assess and advance a model. Considerable effort has been made since the early development of Meso-NH to provide extensive evaluations. Here, we give a comprehensive review of such efforts in the frameworks of intercomparison exercises, field campaigns, and other specific contexts.





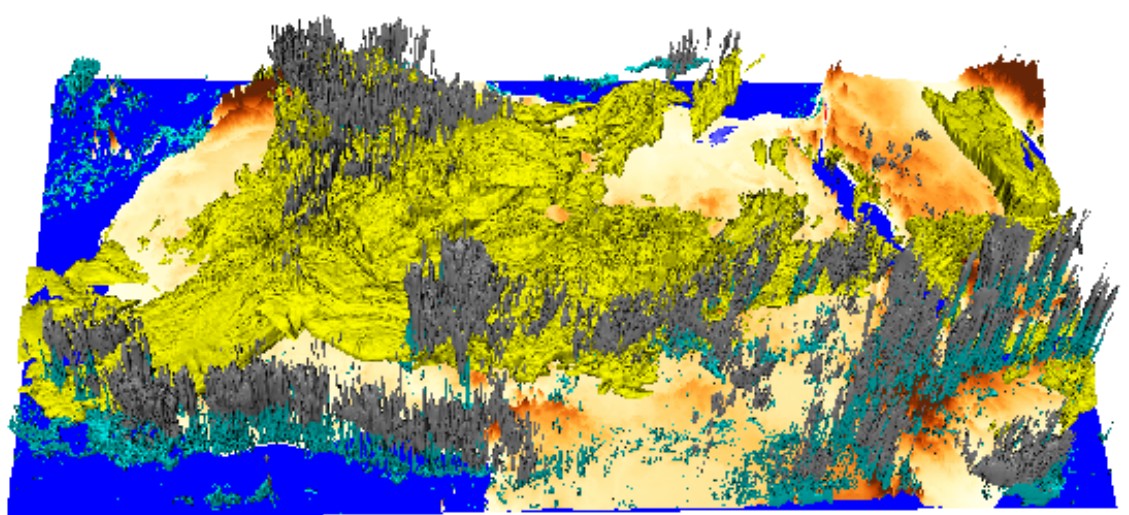

**Figure 16.** Cloud (gray), precipitation (blue), and dust (yellow) during a dusty outbreak at 12:00 UTC on 12 June 2006 over the northern part of Africa as obtained by a simulation using a grid of $3072 \times 1536 \times 70$ points and a grid mesh of $\Delta x = 2.5 \, \mathrm{km}$.

## 8.1 Intercomparison exercises

Meso-NH has joined multiple intercomparison studies to compare state-of-the-art SCM, CRM, or LES simulations with observations and with each other to determine the strengths and weaknesses of the parameterizations. Numerous studies have evaluated the KFB deep convection scheme in the 1D configuration and intercompared it with other schemes/models (e.g.,

Mallet et al., 1999; Bechtold et al., 2000; Xie et al., 2002; Bechtold et al., 2004; Guichard et al., 2004; Woolnough et al., 2010; Couvreux et al., 2015). Initiated under the Global Energy and Water Cycle Experiment (GEWEX) project with the GEWEX Cloud System Study (GCSS) working group (Bechtold et al., 2000; Redelsperger et al., 2000; Stevens et al., 2001; Xie et al., 2002), many of these intercomparison studies involving Meso-NH have focused on deep and boundary layer clouds (Siebesma et al., 2003; Lenderink et al., 2004; Guichard et al., 2004; Woolnough et al., 2010; Varble et al., 2011; Fridlind et al., 2012;

Varble et al., 2014a, b; Daleu et al., 2016a, b; Field et al., 2017). These studies have allowed progress in convection parameterizations and microphysical schemes. In Varble et al. (2011), Meso-NH simulations with one-moment and two-moment microphysics presented convective radar reflectivities closer to the observations than did other models. In some studies, Meso-NH was used as a reference LES simulation to compare to SCM models, e.g., Couvreux et al. (2015), after evaluating the LES against numerous observations and verifying that it correctly reproduced the growth of the boundary layer, development

of shallow cumulus, and initiation of the observed deep convection in a semi-arid environment. Following the GEWEX Atmospheric Boundary Layers Study (GABLS), Cuxart et al. (2006); Bravo et al. (2008); Svensson et al. (2011) focused on the





stable boundary layer; its parameterization is a difficult issue, resulting in a large spread in the intercomparison results. Bergot et al. (2007) intercompared SCM predictions of radiation fog, and Meso-NH overpredicted the cloud water content because the model did not include droplet sedimentation, which has been introduced since then because it is crucial to fog prediction. Other useful intercomparison studies have investigated flow over sloped terrain (Doyle et al., 2000; Georgelin et al., 2000)

and midlatitude and Mediterranean precipitating cloud systems (Lopez et al., 2003; Richard et al., 2003; Anquetin et al., 2005; Barthlott et al., 2011; Khodayar et al., 2016). Several intercomparisons have also examined the dispersion and chemistry (Barth et al., 2007b; Sarrat et al., 2007a; Berger et al., 2016a). In the Fennec dust forecast intercomparison over the Sahara in June 2011 (Chaboureau et al., 2016), Meso-NH at 5 km grid spacing was the only model to partly forecast the large near-surface dust concentration generated by the density current and low-level winds observed by the airborne LIDAR.

## 8.2 Field campaign evaluations

Measurements of atmospheric fields from intensive campaigns at specific locations of interest and for limited time periods are an important source of data used to evaluate Meso-NH. Of the more recent campaigns, the modeling of clouds and convection has been extensively evaluated during the African Monsoon Multidisciplinary Analysis (AMMA) (e.g., Arnault and Roux, 2010; Couvreux et al., 2012), the Convective and Orographically induced Precipitation Study (COPS) (Richard et al., 2011),

CHUVA (Machado et al., 2014), and HyMeX (e.g., Defer et al., 2015; Bouin et al., 2017). Stable boundary layer schemes have benefited from SABLES98 (Cuxart et al., 2000b). Numerous campaigns dedicated to air quality, such as ESCOMPTE (Drobinski et al., 2007), EUCAARI (Aouizerats et al., 2010; Bègue et al., 2015), and CAPITOUL (Masson et al., 2008), have allowed the aerosol and chemistry schemes to be improved and evaluated.

The model has also been used to deliver real-time forecasts to help guide aircraft during several field campaigns. Due to the

limited computer resources of more than 12 years ago, the model was run over a small, coarse grid in 2004 and 2005 for the Tropical Convection, Cirrus, and Nitrogen Oxides experiment (Chaboureau and Bechtold, 2005; Chaboureau and Pinty, 2006) and in 2006 for AMMA (Söhne et al., 2008). Because the computing capability increased at LA after 2007, the model was then run over a larger, finer grid and in the convection-permitting mode for COPS (Chaboureau et al., 2011), Fennec (Chaboureau et al., 2016), HyMeX (Rysman et al., 2016), and CHUVA (Machado and Chaboureau, 2015). Forecasts were produced for

a typical period of one or two months. This provided a long series of simulations compared to the single case simulations commonly done in the past for one or two days. The assessment of such long series against satellite observations has revealed systematic errors or drawbacks in the model. This has led to the development of the subgrid cloud scheme (Chaboureau and Bechtold, 2005) and to the introduction of a temperature dependence in the ice-to-snow autoconversion threshold (Chaboureau and Pinty, 2006).

## 8.3 Other systematic evaluations

As described earlier, since 2008, Meso-NH physics has been used in the operational model AROME and has benefited from systematic evaluations based on the French operational observation network and the forecasters assessment. The performance of Meso-NH has also been evaluated on sites offering long-term statistics, such as the optical turbulence applied to ground-



based astronomy for a statistically rich sample of nights above the European Southern Observatory (ESO) sites (Lascaux et al., 2013; Masciadri et al., 2013, 2017), Arizona (Turchi et al., 2017), and Antarctica (Lascaux et al., 2010, 2011).

Another important aspect is that new versions and bug fixes of the code are systematically validated with a series of test cases including numerous diagnostics covering a wide range of settings, from idealized scenarios including linear mountain

waves (compared to the analytic solution), a density current test case, convective supercells with chemistry and electricity, 1D simulations, and LES intercomparison cases of cumulus, stratocumulus, and fog to real cases of heavy precipitating events, dust outbreaks, and tropical cyclones. This process is handled rigorously because it is critical to maintain consistency in the code. A few of these test cases are provided within the Meso-NH package.

## 9   Future plans

Even though it is complete as an atmospheric model and enables various innovative applications, Meso-NH is continually being developed. Future directions primarily concern computational adaptations, dynamics, physical parameterizations, and integrated coupling systems.

The coming years will see continued work on the computational performance of the code. As in most meteorological codes, all operations are executed in 64-bit double precision even though this is only required for some precision-sensitive operations,

such as the pressure solver. A gain in performance has been achieved by running the model in 32-bit single precision, instead of double. Preliminary tests have been done running the model with single precision but computing the radiative transfer and solving the pressure with double precision. These tests show no strong impact on the accuracy and have a computational cost that is reduced by approximately a factor of 2.

Meso-NH will continue to be adapted to new generations of supercomputers. The next-generation exaflop supercomputers

capable of $10^{18}$ operations per second will be CPU/GPU hybrid machines. OpenACC directives are currently being incorporated into the code and the work done so far results in an acceleration of the advection and turbulence schemes by a factor of 20, reducing the total computational cost by a factor of 3.

Regarding the dynamics, Kurowski et al. (2014) have shown that the choice of anelastic or fully compressible equations is less crucial than the accuracy of the numerical methods. Nevertheless, in some conditions, such as steep slopes where

the pressure solver fails to converge or when horizontal density fluctuations cannot be neglected (in very large domains or in the vicinity of a fire front), the anelastic assumption could become a strong limitation. A compressible version of Meso-NH associated with an adequate time-splitting method will therefore be implemented instead of the anelastic version. This enhancement will allow for the better representation of fire-induced gusts and the strong convection, which are responsible for extreme fire behavior, in particular, the emission and transport of embers or fire jumps.

To improve LESs over complex terrains or with strong surface heterogeneities, such as urban areas, or to study wind turbine emplacements, the drag approach already developed in Meso-NH (Aumond et al., 2013) will not be sufficient. An immersed boundary method is currently being developed to progress beyond the drag approach.





Regarding the physical parameterizations, even though the increasing resolution and the extensive use of LESs reduce the need for some parameterizations, progress in other parameterizations is still needed. It will be necessary to integrate recent advances in radiation schemes, especially for the treatment of 3D radiative effects. Even though ecRad allows subgrid 3D effects to be accounted for, these effects are not considered at the resolved scales, which can be critical in LESs (Klinger et al.,

2017). Cloud side illumination and leakage, horizontal transport between neighbor columns, and cloud shadow projection should therefore be further considered. This will be done by taking advantage of the models recently developed to address these issues (e.g., Wapler and Mayer, 2008; Jakub and Mayer, 2015). The optical properties of clouds should also be revised to benefit from recent theoretical and experimental advancements. In particular, for ice cloud properties, the parameterizations of Liou et al. (2008) for the ice cloud effective radius and that of Yang et al. (2013) for the scattering properties should

be implemented. Regarding aerosols and radiatively active gases, efforts should be made to provide realistic 3D initial and boundary conditions and to improve the prognostic schemes. This will be done within the microphysics scheme LIMA, using analyses from CAMS or MOCAGE. In addition to these structural upgrades, a photovoltaic (PV) module will be implemented to respond to the solar industry's need for improved PV production forecasts.

LESs could benefit from a better representation of turbulence during stable conditions. Another issue is the introduction of

anisotropy in convective clouds or around steep gradients. Parameterizing the turbulence at the cloud–clear air interface is also a difficult and promising challenge that will be dealt with by Meso-NH, depending on the relative magnitudes of the mixing and the phase change time scales and impacting the particle size distributions. Implementing a detailed bin microphysical scheme as a reference for the bulk schemes would allow progress in the parameterization of the microphysical processes.

As shown in this paper, the new couplings developed in Meso-NH, such as aerosols and chemistry, electricity, wildland

fires, oceans, and waves have significantly widened the scope of the applications of the model. It is essential to capitalize on the wealth of these schemes to explore these multiple interactions. An integrated coupled ocean–wave–aerosols–two-moment microphysics–electricity system is a capability that Meso-NH will provide in the near future.

## 10   Outlook

The paper has shown that Meso-NH performs well over a broad range of atmospheric conditions and that many domains of

research can be pursued with Meso-NH version 5.4. In the field of spatial scales, LESs have been extensively used for both process studies and the development of new physical parametrizations for large scale models, and now also concern real case framework, such as over sloping or heterogeneous surfaces. In the near future, LESs will help to improve parametrizations at subkilometric scale as the greyzone of turbulence remains a challenging topic. The new couplings will allow to explore multiple interactions and to access Earth integrating system. In this context, high computational performance of the code used

over large grids represents a genuine challenge.

Over the years, Meso-NH has unified a research community around atmospheric mesoscale modeling and has enabled new challenging issues to be raised. Even though its development began in the 1990s, it has succeeded in integrating new numerical schemes and physical parameterizations, adapting to new computing generations, and remaining attractive for new couplings.

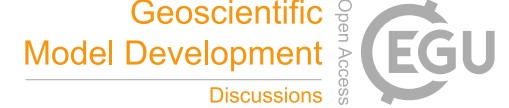



Looking to the future, Meso-NH will continue to serve the atmospheric community, with an increasingly important role for LESs and large grid simulations. Even though global models now have access to kilometric resolutions, limited-area models will remain unavoidable to make progress in the development of parameterizations and the understanding of physical and chemical processes. Meso-NH will remain a major multidisciplinary player.

## 11   Code and data availability

Since version 5.1 was released in 2014, Meso-NH has been freely available under the CeCILL-C license agreement. CeCILL is a free software license, explicitly compatible with GNU GPL. The CeCILL-C license agreement grants users the right to modify and re-use the covered software. The Meso-NH software and its documentation can be downloaded at http://mesonh.aero.obs-mip.fr.

## List of Figures

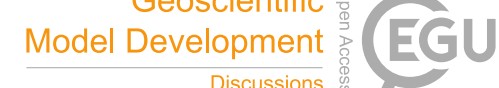







**List of Tables**

*Acknowledgements.* This work was partly supported by the French ANR-14-CE01-0014 MUSIC project. Computer resources were allocated by GENCI (project 0569).



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
