# Peer review of "Overview of the Meso-NH model version 5.4 and its applications"

_Geoscientific Model Development, 2017_

## Referee Comment (RC1) · Anonymous Referee #1 · 13 Feb 2018

General comments :

The paper aims at gathering Meso-NH community model current status and recent developments since the general paper of Lafore et al. (1998) and will constitute an important reference for many users and for the future studies. A large number of developments and studies has been made available over the last 20 years but are summarized as regards the most recent scientific results. The paper presents Meso-NH components starting from the oldest components to the recent additional capabilities, from the core (dynamic), the physics, chemistry, diagnostics, couplings and evaluation. Although the scientific content to be covered is huge, the authors succeeded to make it relatively easy to read and to allow readers finding details on the atmospheric processes of interest but by being aware of the links between the different components

and available tools.

The paper is well-witten and concise. Some clarifications listed below could however help improving the paper.

Due to the large number of schemes and their dependencies, I would suggest that the authors could add a table (or a figure) summarizing the available options (scheme's name + main reference or section) for a process and the links between the schemes (it could replace or augment Table 4, which I think is not meaningful enough).

for example (as I understood the links), microphysics –> ICE3 (single-moment) –> ICE4 (hail) –> CELLS (electricity and lightning)

/ or LIMA (double-moment)

and if there are retroactions (coupling) surface –> SURFEX <–> water <—> NEMO <—> sea salt emissions

/ or CROCO

for example, some additional links could be clarified. As stated in Table 4, there is some atmospheric chemistry research regarding electricity. I understand it is one-way coupling but it is not mentioned elsewhere.

Specific comments:

3.4 numerical diffusion | p.9, l.6. precise if it is CEN4TH.

3.6 Initial and boundary conditions | p.11, l.25: ceiling : are there some considerations to use above conditions from the LS grid instead of using an absorbing layer ?

4.1 surface | p.12, l.12: refer to section 7 for the use of the interface. p.13, l.9: you could name it slab instead of big leaf, which is commonly used for this type of model p.13, l.16- l.18: this sentence could be rewritten ...the TEB scheme approximates the real city 3D structure by resuming this landscape in the form of an urban canyon. ... p.13,

l.21: 'due to the larger surface in contact with the atmosphere' : please add: ... and to the city materials with large heat capacities... p.13, l.27: Is ice only considered over inland water ? Are glaciers considered as part of land surface processes with ISBA ? What about sea ice ? p.13, l.29: is it through a simple aerodynamic roughness length parametrization ? p.14, l.1: how was the 300-m urban local climate zones database created ?

4.2 turbulence | Some clarifications needed. Is it the user who specifies T1D or T3D ? Or is it depending on the grid spacing (T3D below grid spacings of 2 km ) ? Is it the user who specifies mesoscale or LES ? Or is it depending on the grid spacing (LES automatic below grid spacings of 500 m) ? Are there clear recommendations from Meso-NH community experience or is it still an area of investigation ?

4.3 convection and dry thermals | please clarify. -p16, l. 13: The first statement is confusing it should be clarified. It says that shallow and deep convective clouds parametrization is needed for grid spacings larger than 5 km, but latter in the text it is stated that shallow convection with PMMC09 improves clouds up to 500 m- 1 km. So the authors recommend it for small grid spacings ? -p.16, l. 28 : the name PMMC09 is provided too late in the section. -p.17, l. 4: are those modifications to the grey zone already some options available for the users or is it still under investigation ?

4.7 electricity | p. 21, l. 22 / ICE4 is not mentioned in the microphysics section 4.4. Is it an extension developed only for electricity ? If not, it should be presented in section 4.4.

As this component do not appear in figure 2, it could be a sub-section of the microphysics section.

5.1 emissions and dry deposition | p. 23, l. 18 / mention that a more detailed presentation of coupling over water is provided in section 7.1

7.5 Chemistry and aerosols | p.37, l.8-9: "The SO2 concentration modelled for the

plume is close to the observations". I believe the authors, but it is hard to see it in figure 14, we don't see rings colours for the aircraft location (or is it because the colours are the same than the background?)

10 outlook | p.43, 27: the sentence "in the near future..." would better be in section 9

Technical corrections:

table 4 / Turbulence: weather process studies; and Electricity: weather AND process studies ?

References Barthe et al. 2012A and 2012b are the same

7.2.1 urban studies p/32, l.16: replacing building by developing is preferred for this section

p.42, l.23 and p.43, l.1 : repetition of regarding, please change one of the sentences.

---

## Referee Comment (RC2) · Anonymous Referee #2 · 12 Mar 2018

The authors have done a very decent job of summarising the Meso-NH model's many configurations and applications. It appears to be very thorough, is written clearly and reasonably easy to follow. I just have a few minor points of clarification:

1) section 2.1: with the two-way interactive nesting, what frequency of updating do you typically use both to provide the boundary forcing for the "son" and also the upscale relaxation for the "father"? These details should be given in the examples cited later in the paper.

2) section 3.6: are there any issues when nesting this anelastic model inside an NWP model (such as ARPEGE) than uses a different equation set? For example, is it even possible to match completely the temperature, pressure, height and density profiles? Also, how do you choose the reference profile that is needed under the anelastic ap-

proximation in these cases?

3) section 4.3: this section is slightly confusing in that it opens with "The convection scheme available in Meso-NH is KFB..." but then goes on to say there is in fact another, preferred scheme, PMMC09. It would be much clearer to say at the outset how many schemes are available and then to be clear too about which scheme is preferred in what configuration (be it resolution or application)
* * *

---

## Author Comment (AC1) · 9 Apr 2018

Answer to Reviewer 1 : gmd-2017-297-RC1

We thank Referee 1 for his/her comments. We answered below to all the points. Changes made to the original version of the paper appear in track-change mode on the enclosed pdf.

Ref 1:

Due to the large number of schemes and their dependencies, I would suggest that the authors could add a table (or a figure) summarizing the available options (scheme's name + main reference or section) for a process and the links between the schemes (it could replace or augment Table 4, which I think is not meaningful enough). for

example (as I understood the links), microphysics –> ICE3 (single-moment) –> ICE4 (hail) –> CELLS (electricity and lightning) / or LIMA (double-moment) and if there are retroactions (coupling) surface –> SURFEX <–> water <—> NEMO <—> sea salt emissions / or CROCO for example, some additional links could be clarified.

Authors: We agree that additional information is necessary to clarify the links between the schemes. Table 4 was previously asked by the Editor, it has been completed with some information about the schemes and their links. Also, a new figure (Fig.6) has been added to show the one-way or two-way links between the schemes.

Ref 1: As stated in Table 4, there is some atmospheric chemistry research regarding electricity. I understand it is one-way coupling but it is not mentioned elsewhere.

Authors: You are right, a sentence has been added in Part 4.7Âă: A lightning-produced NOx (LNOx) parameterization is implemented in the electrical scheme. Since the CELLS scheme reproduces the lightning flash path, the LNOx production is taken proportional to the lightning flash length and depends on the atmospheric pressure (Barthe et al., 2007).Âă

Ref 1 Specific comments: 3.4 numerical diffusion | p.9, l.6. precise if it is CEN4TH.

Authors: Yes, it is.

Ref 1: 3.6 Initial and boundary conditions | p.11, l.25: ceiling : are there some considerations to use above conditions from the LS grid instead of using an absorbing layer ?

Authors: The absorbing layer uses LS fields to relaxe prognostic variables towards them.

Ref 1: 4.1 surface | p.12, l.12: refer to section 7 for the use of the interface.

Authors: The introduction of section 7 has been clarified as the coupling interface in SURFEX exists for all the schemes, and has allowed the coupling with 3D ocean

models.

Ref 1: p.13, l.9: you could name it slab instead of big leaf, which is commonly used for this type of model

Authors: All right, done.

Ref 1: p.13, l.16- l.18: this sentence could be rewritten ...the TEB scheme approximates the real city 3D structure by resuming this landscape in the form of an urban canyon. ...

Authors: Thank you

Ref 1: p.13, l.21: 'due to the larger surface in contact with the atmosphere' : please add: ... and to the city materials with large heat capacities...

Authors: Thank you

Ref 1: p.13, l.27: Is ice only considered over inland water ? Are glaciers considered as part of land surface processes with ISBA ? What about sea ice ?

Authors: Permanent snow is treated in the ISBA scheme as very deep snow. Sea ice is treated either where SST temperature is below -4°C or by the s GELATO ea ice model (Mélia, 2002) coupled with the 3D ocean model. These elements have been added.

Ref 1: p.13, l.29: is it through a simple aerodynamic roughness length parametrization ?

Authors: No, the fluxes are directly simulated, using a statistical fit coming from various experimental campaigns (Belamari and Pirani, 2007). This reference has been added in the text. Belamari, S. and Pirani, A.: Validation of the optimal heat and momentum fluxes using the ORCA-LIM global ocean-ice model, MERSEA IP Deliverable, D.4.1.3, 88 pp., 2007.

Ref 1: p.14, l.1: how was the 300-m urban local climate zones database created ?

Authors: The urban LCZ were derived from the global human Settlement Layer produced by JRC: Pesaresi M., Guo H., Blaes X., Ehrlich D., Ferri S., Gueguen L., Halkia M., Kauffmann M., Kemper T., Lu L., Marin-Herrera M.A., Ouzounis G.K., Scavazzon M., Soille P., Syrris V. and L. Zanchetta A Global Human Settlement Layer From Optical HR/VHR RS Data: Concept and First Results. IEEE J. Sel. Top. Appl. Earth Obs. Remote Sens. 6(5):2102–2131, 2013. doi:10.1109/JSTARS.2013.2271445.

Ref 1: 4.2 turbulence | Some clarifications needed. Is it the user who specifies T1D or T3D? Or is it depending on the grid spacing (T3D below grid spacings of 2 km ) ? Is it the user who specifies mesoscale or LES ? Or is it depending on the grid spacing (LES automatic below grid spacings of 500 m) ? Are there clear recommendations from Meso-NH community experience or is it still an area of investigation ?

Authors: T1D or T3D, determining mesoscale or LES mode, and the mixing length parametrization are chosen by the user according to clear recommendations given above. This remark has been added.

Ref 1: 4.3 convection and dry thermals | please clarify. -p16, l. 13: The first statement is confusing it should be clarified. It says that shallow and deep convective clouds parametrization is needed for grid spacings larger than 5 km, but latter in the text it is stated that shallow convection with PMMC09 improves clouds up to 500 m- 1 km. So the authors recommend it for small grid spacings ? -p.16, l. 28 : the name PMMC09 is provided too late in the section.

Authors: Clarification has been brought.

Ref 1: -p.17, l. 4: are those modifications to the grey zone already some options available for the users or is it still under investigation ?

Authors: These options are available in version 5.4, but the question is still under investigation.

Ref 1: 4.7 electricity | p. 21, l. 22 / ICE4 is not mentioned in the microphysics section

4.4. Is it an extension developed only for electricity ? If not, it should be presented in section 4.4. As this component do not appear in figure 2, it could be a sub-section of the microphysics section.

Authors: Thank you, the introduction of ICE4 in the microphysics was missing, as ICE4 does not exist only for electricity. This has been also clarified at different locations, including the microphysics figure caption.

Ref 1: 5.1 emissions and dry deposition | p. 23, l. 18 / mention that a more detailed presentation of coupling over water is provided in section 7.1

Authors: Yes, thank you.

Ref 1: 7.5 Chemistry and aerosols | p.37, l.8-9: "The SO2 concentration modelled for the plume is close to the observations". I believe the authors, but it is hard to see it in figure 14, we don't see rings colours for the aircraft location (or is it because the colours are the same than the background?)

Authors: You are right that it is hard to see it in Fig.14. A few sentences have been deleted.

Ref 1: 10 outlook | p.43, 27: the sentence "in the near future..." would better be in section 9

Authors: You are completely right, thank you.

Ref 1: Technical corrections: table 4 / Turbulence: weather process studies; and Electricity: weather AND process studies ? Authors: Thank you, it has been corrected.

Ref 1: References Barthe et al. 2012A and 2012b are the same Authors: Thank you, it has been corrected.

Ref 1: 7.2.1 urban studies p/32, l.16: replacing building by developing is preferred for this section Authors: Yes, done.
Ref 1: p.42, l.23 and p.43, l.1 : repetition of regarding, please change one of the sentences. Authors: Yes, done.

Thank you very much for the time you have put into the correction of this paper and the relevance of your remarks.

Please also note the supplement to this comment:

[revised manuscript text omitted]

---

## Author Comment (AC2) · 9 Apr 2018

We thank Referee 2 for his/her comments. We answered below to all the points. Changes made to the original version of the paper appear in track-change mode on the enclosed pdf.

Ref2: 1) section 2.1: with the two-way interactive nesting, what frequency of updating do you typically use both to provide the boundary forcing for the "son" and also the upscale relaxation for the "father"? These details should be given in the examples cited later in the paper.

Authors: Spatial interpolating is performed only when the two models are synchronized in time. So the exchange of inforWe thank Referee 2 for his/her comments. We answered below to all the points. Changes made to the original version of the paper appear in track-change mode on the enclosed pdf.

Ref2: 1) section 2.1: with the two-way interactive nesting, what frequency of updating do you typically use both to provide the boundary forcing for the "son" and also the upscale relaxation for the "father"? These details should be given in the examples cited later in the paper.

Authors: Spatial interpolating is performed only when the two models are synchronized in time. So the exchange of information between the nested models occurs at each coarse mesh model time step, as illustrated in the joined figure from Stein et al. (2000). This has been added in the text.

Ref 2: 2) section 3.6: are there any issues when nesting this anelastic model inside an NWP model (such as ARPEGE) than uses a different equation set? For example, is it even possible to match completely the temperature, pressure, height and density profiles? Also, how do you choose the reference profile that is needed under the anelastic approximation in these cases?

Authors: There is probably a confusion in the sense that there is no nesting between Meso-NH and the NWP model. This probably comes from the sentence: initial and coupling fields can be provided by analyses or forecasts from the following NWP suites. The term coupling is replaced by forcing as there is no feedback from Meso-NH to the NWP model. There is no issue to initialize and force Meso-NH with a coarse model presenting different governed equations. At the initialization, thermodynamical fields are first adapted to the Meso-NH variables (absolute temperature to virtual potential temperature, specific humidity to vapor mixing ratio). Then pressure, potential temperature and mixing ratio are interpolated to the new grid. The reference state is computed from the virtual potential temperature, the mixing ratio and the reference state Exner function at model top, using the hydrostatic equation. Wind fields are then interpolated, and the anelastic balance corrects them to get a final non-divergent wind field.
Ref 2: 3) section 4.3: this section is slightly confusing in that it opens with "The convection scheme available in Meso-NH is KFB..." but then goes on to say there is in fact another, preferred scheme, PMMC09. It would be much clearer to say at the outset how many schemes are available and then to be clear too about which scheme is preferred in what configuration (be it resolution or application).

Authors: You are absolutely right and this has been corrected.mation between the nested models occurs at each coarse mesh model time step, as illustrated in the joined figure from Stein et al. (2000). This has been added in the text.

Ref 2: 2) section 3.6: are there any issues when nesting this anelastic model inside an NWP model (such as ARPEGE) than uses a different equation set? For example, is it even possible to match completely the temperature, pressure, height and density profiles? Also, how do you choose the reference profile that is needed under the anelastic approximation in these cases?

Authors: There is probably a confusion in the sense that there is no nesting between Meso-NH and the NWP model. This probably comes from the sentence: initial and coupling fields can be provided by analyses or forecasts from the following NWP suites. The term coupling is replaced by forcing as there is no feedback from Meso-NH to the NWP model. There is no issue to initialize and force Meso-NH with a coarse model presenting different governed equations. At the initialization, thermodynamical fields are first adapted to the Meso-NH variables (absolute temperature to virtual potential temperature, specific humidity to vapor mixing ratio). Then pressure, potential temperature and mixing ratio are interpolated to the new grid. The reference state is computed from the virtual potential temperature, the mixing ratio and the reference state Exner function at model top, using the hydrostatic equation. Wind fields are then interpolated, and the anelastic balance corrects them to get a final non-divergent wind field.

Ref 2: 3) section 4.3: this section is slightly confusing in that it opens with "The convection scheme available in Meso-NH is KFB..." but then goes on to say there is in

fact another, preferred scheme, PMMC09. It would be much clearer to say at the out-set how many schemes are available and then to be clear too about which scheme is preferred in what configuration (be it resolution or application).

Authors: You are absolutely right and this has been corrected.

Please also note the supplement to this comment:
https://www.geosci-model-dev-discuss.net/gmd-2017-297/gmd-2017-297-AC2-supplement.pdf
* * *
[Figure]

[Figure]

[Figure]

Fig. 1. Schematic diagram for the nesting configuration. The upper panel gives an example of the spatial distribution for the 4 models and the lower panel shows the nesting of the time-steps

**Fig. 1.**

**Supplement:**

[revised manuscript text omitted]

---

## Author Response (AR1)

**We thank Referee 1 for his/her comments. We answered below to all the points. The comments are in italic font while our answers appear in blue normal font. Changes made to the original version of the paper appear in track-change mode on the enclosed pdf.**

*Due to the large number of schemes and their dependencies, I would suggest that the authors could add a table (or a figure) summarizing the available options (scheme's name + main reference or section) for a process and the links between the schemes (it could replace or augment Table 4, which I think is not meaningful enough). for example (as I understood the links), microphysics –> ICE3 (single-moment) –> ICE4 (hail) –> CELLS (electricity and lightning)*
*/ or LIMA (double-moment)*
*and if there are retroactions (coupling) surface –> SURFEX <–> water <—> NEMO*
*<—> sea salt emissions*
*/ or CROCO*
*for example, some additional links could be clarified.*

We agree that additional information is necessary to clarify the links between the schemes. Table 4 was previously asked by the Editor, it has been completed with some information about the schemes and their links. Also, a new figure (Fig.6) has been added to show the one-way or two-way links between the schemes.

*As stated in Table 4, there is some atmospheric chemistry research regarding electricity. I understand it is one-way coupling but it is not mentioned elsewhere.*

You are right, a sentence has been added in Part 4.7 : A lightning-produced NOx (LNOx) parameterization is implemented in the electrical scheme. Since the CELLS scheme reproduces the lightning flash path, the LNOx production is taken proportional to the lightning flash length and depends on the atmospheric pressure (Barthe et al., 2007).

*Specific comments:*
*3.4 numerical diffusion | p.9, l.6. precise if it is CEN4TH.*

Yes, it is.

*3.6 Initial and boundary conditions | p.11, l.25: ceiling : are there some considerations*
*to use above conditions from the LS grid instead of using an absorbing layer ?*

The absorbing layer uses LS fields to relaxe prognostic variables towards them.

*4.1 surface | p.12, l.12: refer to section 7 for the use of the interface.*

The introduction of section 7 has been clarified as the coupling interface in SURFEX exists for all the schemes, and has allowed the coupling with 3D ocean models.

*p.13, l.9: you could name it slab instead of big leaf, which is commonly used for this type of model*

All right, done.

*p.13, l.16- l.18: this sentence could be rewritten ...the TEB scheme approximates the real city 3D structure by resuming this landscape in the form of an urban canyon. ...*

Thank you

*p.13, l.21: 'due to the larger surface in contact with the atmosphere' : please add: ... and to the city materials with large heat capacities...*

Thank you

*p.13, l.27: Is ice only considered over inland water ? Are glaciers considered as part of land surface processes with ISBA ? What about sea ice ?*

Permanent snow is treated in the ISBA scheme as very deep snow. Sea ice is treated either where SST temperature is below -4°C or by the s GELATO ea ice model (Mélia, 2002) coupled with the 3D ocean model. These elements have been added.

*p.13, l.29: is it through a simple aerodynamic roughness length parametrization ?*

No, the fluxes are directly simulated, using a statistical fit coming from various experimental campaigns (Belamari and Pirani, 2007). This reference has been added in the text.
Belamari, S. and Pirani, A.: Validation of the optimal heat and momentum fluxes using the ORCA-LIM global ocean-ice model, MERSEA IP Deliverable, D.4.1.3, 88 pp., 2007.

*p.14, l.1: how was the 300-m urban local climate zones database created ?*

The urban LCZ were derived from the global human Settlement Layer produced by JRC:
Pesaresi M., Guo H., Blaes X., Ehrlich D., Ferri S., Gueguen L., Halkia M., Kauffmann M., Kemper T., Lu L., Marin-Herrera M.A., Ouzounis G.K., Scavazzon M., Soille P., Syrris V. and L. Zanchetta A Global Human Settlement Layer From Optical HR/VHR RS Data: Concept and First Results. IEEE J. Sel. Top. Appl. Earth Obs. Remote Sens. 6(5):2102–2131, 2013. doi:10.1109/JSTARS.2013.2271445.

*4.2 turbulence | Some clarifications needed. Is it the user who specifies T1D or T3D? Or is it depending on the grid spacing (T3D below grid spacings of 2 km ) ? Is it the user who specifies mesoscale or LES ? Or is it depending on the grid spacing (LES automatic below grid spacings of 500 m) ? Are there clear recommendations from Meso-NH community experience or is it still an area of investigation ?*

T1D or T3D, determining mesoscale or LES mode, and the mixing length parametrization are chosen by the user according to clear recommendations given above. This remark has been added.

*4.3 convection and dry thermals | please clarify. -p16, l. 13: The first statement is confusing it should be clarified. It says that shallow and deep convective clouds parametrization is needed for grid spacings larger than 5 km, but latter in the text it is stated that shallow convection with PMMC09 improves clouds up to 500 m- 1 km. So the authors recommend it for small grid spacings ? -p.16, l. 28 : the name PMMC09 is provided too late in the section.*

Clarification has been brought.

*-p.17, l. 4: are those modifications to the grey zone already some options available for the users or is it still under investigation ?*

These options are available in version 5.4, but the question is still under investigation.

*4.7 electricity | p. 21, l. 22 / ICE4 is not mentioned in the microphysics section 4.4. Is it an extension developed only for electricity ? If not, it should be presented in section 4.4. As this component do not appear in figure 2, it could be a sub-section of the microphysics section.*

Thank you, the introduction of ICE4 in the microphysics was missing, as ICE4 does not exist only for electricity. This has been also clarified at different locations, including the microphysics figure caption.

*5.1 emissions and dry deposition | p. 23, l. 18 / mention that a more detailed presentation of coupling over water is provided in section 7.1*

Yes, thank you.

*7.5 Chemistry and aerosols | p.37, l.8-9: "The SO2 concentration modelled for the plume is close to the observations". I believe the authors, but it is hard to see it in figure 14, we don't see rings colours for the aircraft location (or is it because the colours are the same than the background?)*

You are right that it is hard to see it in Fig.14. A few sentences have been deleted.

*10 outlook | p.43, 27: the sentence "in the near future..." would better be in section 9*

You are completely right, thank you.

*Technical corrections:*
*table 4 / Turbulence: weather process studies; and Electricity: weather AND process studies ?*
Thank you, it has been corrected.

*References Barthe et al. 2012A and 2012b are the same*
Thank you, it has been corrected.

*7.2.1 urban studies p/32, l.16: replacing building by developing is preferred for this section*
Yes, done.

*p.42, l.23 and p.43, l.1 : repetition of regarding, please change one of the sentences.*
Yes, done.

**Thank you very much for the time you have put into the correction of this paper and the relevance of your remarks.**

**We thank Referee 2 for his/her comments. We answered below to all the points. The comments are in italic font while our answers appear in blue normal font. Changes made to the original version of the paper appear in track-change mode on the enclosed pdf.**

*The authors have done a very decent job of summarising the Meso-NH model's many configurations and applications. It appears to be very thorough, is written clearly and reasonably easy to follow. I just have a few minor points of clarification:*

*1) section 2.1: with the two-way interactive nesting, what frequency of updating do you typically use both to provide the boundary forcing for the "son" and also the upscale relaxation for the "father"? These details should be given in the examples cited later in the paper.*

Spatial interpolating is performed only when the two models are synchronized in time. So the exchange of information between the nested models occurs at each coarse mesh model time step, as illustrated in the figure below from Stein et al. (2000). This has been added in the text.

[Figure]

[Figure]

Fig. 1. Schematic diagram for the nesting configuration. The upper panel gives an example of the spatial distribution for the 4 models and the lower panel shows the nesting of the time-steps

*2) section 3.6: are there any issues when nesting this anelastic model inside an NWP model (such as ARPEGE) than uses a different equation set? For example, is it even possible to match completely the temperature, pressure, height and density profiles?*
*Also, how do you choose the reference profile that is needed under the anelastic approximation in these cases?*

There is probably a confusion in the sense that there is no nesting between Meso-NH and the NWP model. This probably comes from the sentence :  «initial and coupling fields can be provided by analyses or forecasts from the following NWP suites ». The term « coupling » is replaced by « forcing » as there is no feedback from Meso-NH to the NWP model.
There is no issue to initialize and force Meso-NH with a coarse model presenting different governed equations. At the initialization, thermodynamical fields are first adapted to the Meso-NH variables (absolute temperature $\rightarrow$ virtual potential temperature, specific humidity $\rightarrow$ vapor mixing ratio). Then pressure, potential temperature and mixing ratio are interpolated to the new grid. The reference state is computed from the virtual potential temperature, the mixing ratio and the reference state Exner function at model top, using the hydrostatic equation. Wind fields are then interpolated, and the anelastic balance corrects them to get a final non-divergent wind field.

*3) section 4.3: this section is slightly confusing in that it opens with "The convection scheme available in Meso-NH is KFB..." but then goes on to say there is in fact another, preferred scheme, PMMC09. It would be much clearer to say at the outset how many schemes are available and then to be clear too about which scheme is preferred in what configuration (be it resolution or application).*

You are absolutely right and this has been corrected.

**Thank you very much for the time you have put into the correction of this paper and the relevance of your remarks.**

[revised manuscript text omitted]